# Lactic Acid: A Comprehensive Review of Production to Purification

**Abidemi Oluranti Ojo * and Olga de Smidt**

Centre for Applied Food Sustainability and Biotechnology (CAFSaB), Central University of Technology, Bloemfontein 9300, South Africa
* Correspondence: aojo@cut.ac.za

**Abstract:** Lactic acid (LA) has broad applications in the food, chemical, pharmaceutical, and cosmetics industries. LA production demand rises due to the increasing demand for polylactic acid since LA is a precursor for polylactic acid production. Fermentative LA production using renewable resources, such as lignocellulosic materials, reduces greenhouse gas emissions and offers a cheaper alternative feedstock than refined sugars. Suitable pretreatment methods must be selected to minimize LA cost production, as the successful hydrolysis of lignocellulose results in sugar-rich feedstocks for fermentation. This review broadly focused on fermentative LA production from lignocellulose. Aspects discussed include (i). low-cost materials for fermentative LA production, (ii). pretreatment methods, (iii). enzymatic hydrolysis of cellulose and hemicellulose, (iv). lactic acid-producing microorganisms, including fungi, bacteria, genetically modified microorganisms, and their fermentative pathways, and (v). fermentation modes and methods. Industrial fermentative lactic acid production and purification, difficulties in using lignocellulose in fermentative LA production, and possible strategies to circumvent the challenges were discussed. A promising option for the industrial production and purification of LA that contains enzyme and cell recycling continuous simultaneous saccharification and fermentation coupled with membrane-based separation was proposed. This proposed system can eliminate substrate-, feedback-, and end-product inhibition, thereby increasing LA concentration, productivity, and yield.

**Keywords:** lactic acid; lignocellulose; pretreatments; fermentation; microorganisms

## 1. Introduction

Lactic acid (2-hydroxypropionic acid), a naturally occurring organic acid, was first discovered in sour milk by Scheele in 1780 [1]. Lactic acid (LA) obtained through fermentation by Fremi in 1881 has led to its industrial production [2]. LA can be produced through chemical synthesis or microbial fermentation of sugars derived from renewable resources, such as agricultural waste materials [3]. Lactic acid is harmless and categorized as Generally Recognized As Safe (GRAS) [3,4]. Lactic acid is of biotechnological importance as it is widely used in the cosmetics, food, pharmaceutical, medical, and chemical industries (Figure 1) [5,6].

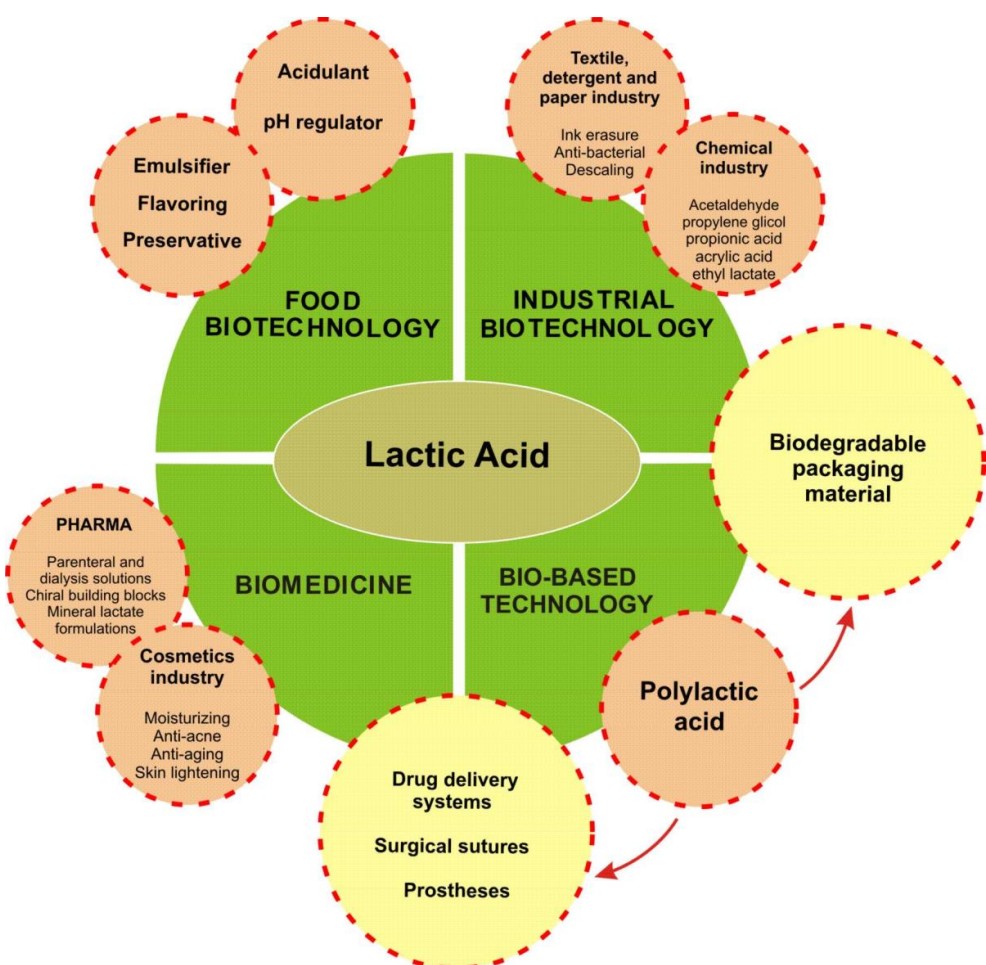

**Figure 1.** A wide range of biotechnological applications of lactic acid (Data from Balla et al. (2021) [7], Jem et al. (2020) [8], Maleki et al. (2022) [9], and Alsaheb et al. (2022) [10]).

Lactic acid has received significant attention; its demand has increased because it is a building block for synthesizing polylactic acid (PLA) [11]. Polylactic acid is eco-friendly and non-toxic; it is an essential polymeric material extensively used for biomedical applications due to its biocompatibility, biodegradability, processability, and mechanical strength [6,12]. PLA can be polymerized into pure poly-D-lactic acid (PDLA) or poly-L-lactic acid (PLLA) and poly-DL-lactic acid (PDLLA) [13]. Lactic acid also serves as a precursor of compounds such as acrylic polymers and propylene glycol used in packaging and labeling [14,15].

Lactic acid ($CH_3CHOHCOOH$) is a chiral molecule in two enantiomeric forms: L-lactic acid and D-lactic acid (Figure 2). Lactic acid can exist in any of its optically active forms (L (+) or D (−)) or in a racemic mixture (L (+) and D (−)), depending on the production processing routes [12,16].

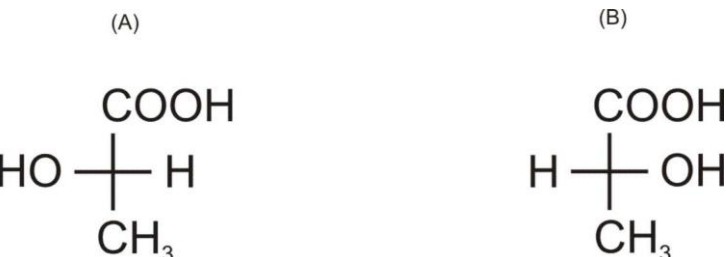

**Figure 2.** Three-dimensional structure of optical L-lactic acid (**A**) and D-lactic acid (**B**) (Adapted from Pohanka (2020) [17]).

Lactic acid is a yellow to colorless liquid (at 15 °C and 1013 bars); it is only soluble in water, ethanol, and other water-soluble miscible organic solvents [18,19]. Due to lactic acid's hygroscopic nature, it is usually obtained as a colorless concentrated solution (up to 90%) [18]. Lactic acid is odorless, less volatile, and the simplest hydroxycarboxylic acid with various physicochemical properties that include melting points of 53.0 °C (L-lactic acid), 52.8 °C (D-lactic acid) and 16.8 °C (racemic LD-lactic acid). Lactic acid boiling points vary at different pressures; for instance, at 1.87 kPa, the boiling point of lactic acid is 103 °C and 122 °C at 1.99 kPa. The solid density of lactic acid at 20 °C is 1.249 g/L, and at 25 °C in the aqueous solution, the density is 1.057 g/mL (for 20% wt.) and 1.201 g/mL (for 88.6% wt.). The lactic acid dissociation constant (p$Ka$) at 25 °C for L and D isomers are 3.79 and 3.83 [18].

The physicochemical properties of lactic acid play an essential role in its chemical behavior. For example, lactic acid exhibits an acidic character in an aqueous medium and has an asymmetric carbon that gives optical activity. It has excellent reaction versatility due to the bifunctional reactivity associated with its carboxyl and hydroxyl groups. In chemical industries, applications of one optically pure lactic acid (L or D) or the mixture are desirable [20]. However, L-lactic acid is preferred in the biomedical and food industries because it can be metabolized by animal cells [20]. In contrast, D-lactic acid cannot be metabolized, and its presence in the body can lead to acidosis [15,16].

## 2. Lactic Acid Production Technologies

The lactic acid annual global market in 2020 was valued at 1.1 billion US dollars and is expected to have a compound annual growth rate (CAGR) of 8% from 2021 to 2028 [21]. Lactic acid usage in end-use industries such as pharmaceuticals, biomedicals, foods, and beverages drives demand over the forecast period [21]. Lactic acid is produced through chemical or microbial fermentative processes [3].

### 2.1. Chemical Synthesis of Lactic Acid Production

Lactic acid is produced by various chemical reactions, including (i) hydrolysis of lactic acid derivatives, e.g., esters or nitriles, (ii) hydrolysis of the other two substituted propionic acids, (iii) decarboxylation of some derivatives of 2-methylmalonic acid, (iv) reduction, (v) oxidation and (vi) rearrangement and disproportion [22]. However, only lactic acid synthesis from its derivatives has been commercialized. Though the chemical synthesis of lactic acid is not economically feasible [23], several studies have reported the chemical synthesis of lactic acid using different carbon sources. For example, lactic acid can be chemically synthesized from a petrochemical source. The reaction steps in lactic acid production using petrochemical sources include the oxidation of ethene in the presence of palladium (II) chloride to form acetaldehyde (Figure 3A). The acetaldehyde in the liquid phase under high pressure with hydrogen cyanide in the presence of a base is converted into lactonitrile. Lactonitrile is recovered and purified, and sulphuric acid is used to hydrolyze lactonitrile to form a racemic mixture of L- and D-lactic acid [24]. The reactions in the synthesis of lactic acid or its lactate involve the addition of glycerol, water, and a catalyst (sodium or potassium hydroxide) into a batch reactor equipped with a magnetic stirrer set at 800 rpm and temperature (240–247 °C; 260–270 °C) (Figure 3B) [25].

Alkaline hydrothermal conversion of glycerol yielded chemical synthesis of a racemic mixture of L- and D-lactic acid [25,26]. Zhou et al. (2010) [27] have shown that lactic acid could also be synthesized from mannitol (C6polyol) through alkaline hydrothermal conditions; however, the lactic acid yield from mannitol was lower than the yield obtained from glucose and glycerine. Other routes of chemical synthesis of lactic acid include oxidation of propylene glycol at a low temperature [28], conversion of propene into α-nitropropionic acid using nitric acid in the presence of oxygen, and α-nitropropionic acid is then hydrolyzed into lactic acid [22].

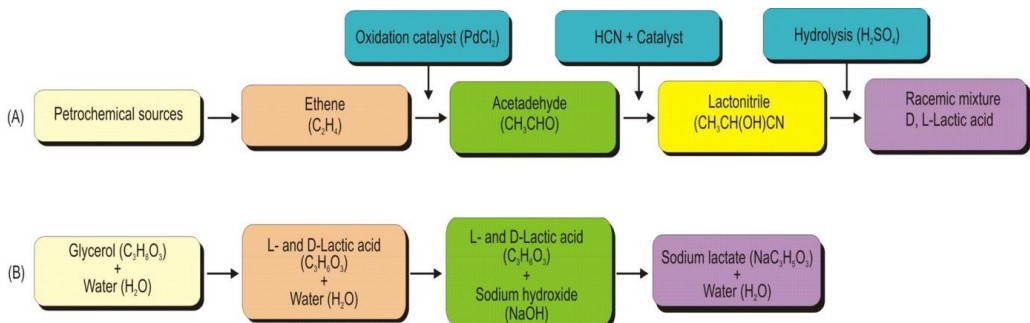

**Figure 3.** Block diagram of chemical synthesis of lactic acid using sources and conditions. (**A**): Synthesis of lactic acid using petrochemical source and (**B**): Alkaline hydrothermal conversion of glycerol to lactic acid and its lactate (Adapted from Yankov (2022) [24]).

*2.2. Fermentative Production of Lactic Acid*

Over 90% of lactic acid is produced through microbial fermentation of carbon sources [25]. In fermentation, fermentable sugars in nutrient supplements are converted to lactic acid by capable microorganisms under favorable conditions [3]. The selection of favorable conditions for fermentative lactic acid production is vital. Factors such as temperature, pH, nutrients, substrate concentration, and end-product concentration, among others, were reported to impact fermentation [29]. The temperature and pH are important conditions in lactic acid fermentation as they are associated with cellular metabolism, which affects the growth of microorganisms, substrate consumption, and lactic acid production [29].

Consequently, the optimum temperature is selected and maintained. The selected optimum pH is maintained by adding a strong base such as calcium hydroxide, sodium hydroxide, or potassium hydroxide during fermentation due to lactic acid production that lowers the pH [30]. Nutrients can influence lactic acid production since microorganisms such as lactic acid-producing bacteria require complex nutrients. The carbon source available as sugars is vital for lactic acid-producing bacteria (LAB) reproduction. Minerals, vitamins, and nitrogen available as inorganic compounds are essential for microbial growth, maintenance, and production of lactic acid [29]. High substrate concentration due to the overloading of carbon sources or the inability of the selected microorganisms to utilize substrates can lead to an inhibition [3,29]. The end-product of the fermentation process could give inhibitory effects because of the accumulation of lactic acid in the system, thereby causing a decrease in cell growth, extended fermentation period, and reduced lactic acid productivity [3,31].

## 3. Low-Cost Raw Materials for Fermentative Lactic Acid Production

The total production cost of lactic acid depends on the starting raw substrate materials, as these materials constitute 40–70% of the total cost of the production [32]. Searching for low-cost raw materials in fermentative lactic acid production has gained more attention since using refined sugars such as glucose and sucrose as feedstock to produce lactic acid are very expensive [33]. According to Dumbrepatil et al. [34], the manufacturing cost can be reduced if waste products containing fermentable sugar are used in lactic acid production. For cost-effectiveness in lactic acid production, the selected cheap raw materials should have the properties to produce high yield, high productivity, less by-product formation, and little or no contamination [11]. The most common low-cost raw materials used in lactic acid production are agricultural wastes and food industry by-products [34–36]. Some food processing by-products and agricultural wastes classified as disaccharides and polymeric substrates [3] are used to produce monosaccharides for fermentative lactic acid production.

*3.1. Disaccharides*

Molasses and whey are the most common low-cost disaccharides food processing by-products used as feedstock for lactic acid production. Molasses, mostly found in sugar beet

and sugarcane plants, is a thick brown syrup left over after removing sugar crystals. The advantage of using cane molasses as a substrate is its high sucrose content [37] which can be hydrolyzed to form monosaccharides (glucose and fructose) for lactic acid production. Molasses contains low concentrations of heavy metals and alkali earth ions that may affect the media pH, inhibit cell growth, and deactivate enzymes involved in product formation; the addition of sulphuric acid to molasses before thermal pretreatment improves fermentation efficiency [36,37]. Additional purification steps may be required depending on the source of molasses, though this may increase lactic acid production costs [33,36,38].

In hydrolysis and thermal pretreatment, sulphuric acid (0.2% final concentration) is used to hydrolyze molasses, followed by thermal treatment at 100 °C for 20 min [34]. Enzymatic pretreatment follows thermal pretreatment; this involves using enzymes such as invertase to catalyze sucrose's hydrolysis into glucose and fructose, which can be used as substrates for fermentation. Yang and Montgomery (2007) [39], Dumbrepatil et al. (2008) [34], Mahato et al. (2021) [40], Vidra et al. (2017) [36], among others, described the processes and conditions for fermentation of the monomeric sugars into lactic acid.

Whey is a liquid by-product formed from the cheese production process, and its disposal has an environmental challenge [41,42]. Whey has a high lactose content [43] and can be hydrolyzed into glucose and galactose (monosaccharides).

### 3.2. Polymeric Substrates

Food waste and lignocellulosic biomass (polymeric substrates) have recently replaced petroleum-based resources in fermentative lactic acid production due to their cost effectiveness, economic sustainability, and environmental preservation [3,6,44,45]. Food waste is usually rich in carbohydrates (e.g., starch) [46]. Starch, a polysaccharide, comprises glucose monomers joined in $\alpha$ 1, 4 linkages forming various lengths (branched or unbranched) [47,48]. Starch has two polysaccharide fractions (amylose and amylopectin). Amylose, the linear polymer, is the simplest form of starch, and amylopectin is the branched form [48]. When microorganisms cannot directly assimilate starch, starch is hydrolyzed into glucose before lactic acid fermentation [3].

Lignocellulosic biomass is the preferred substrate as it meets a huge demand in lactic acid production [49]. Lignocellulosic biomass, a natural carbon source, is an organic material obtained from a biological source, and it represents an unutilized, most abundant global source of biomass [50]. Lignocellulose is the building block of plant cell walls composing cellulose wrapped by the dense structures of hemicellulose and lignin [51,52]. Lignocellulosic biomass is composed of cellulose (insoluble fibers of β-1,4-glucans), hemicellulose (non-cellulosic polysaccharides such as xylan, glucans, and mannans, etc.), lignin (a complex polyphenolic structure), smaller amounts of ash, pectin, extractives, and proteins [52–55].

The long-chain cellulose polymers, strongly associated with each other by hydrogen bonds and van der Waals bonds, allow the cellulose to be packed into microfibrils. These microfibrils are covered by hemicellulose and lignin [56]. Cellulose, hemicellulose, and lignin form almost 90% of dry matter [54,55]. Cellulose and hemicellulose are carbohydrate polymers, while lignin is a non-carbohydrate phenolic polymer [57]. Cellulose, hemicellulose, and lignin are the structural frameworks of the plant cell walls [33,56], and their distribution varies in the different types and parts of plants (Table 1).

**Table 1.** Percentage of cellulose, hemicellulose, and lignin contents in selected agricultural residues and wastes.

| Lignocellulosic Material | Cellulose (%) | Hemicellulose (%) | Lignin (%) |
|---|---|---|---|
| Brewer spent grains [a] | 24.5 | 23.8 | 15.8 |
| Corncob [b] | 45 | 35 | 15 |
| Hardwood stems [b] | 40–55 | 24–40 | 18–25 |
| Softwood stems [b] | 40–50 | 25–35 | 25–35 |
| Newspaper [b] | 40–55 | 25–40 | 18–30 |
| Wastepaper from chemical pulp [b] | 60–70 | 10–20 | 5–10 |

**Table 1.** *Cont.*

| Lignocellulosic Material | Cellulose (%) | Hemicellulose (%) | Lignin (%) |
|---|---|---|---|
| Grasses [b] | 25–40 | 35–50 | 10–30 |
| Switchgrass [b] | 31–45 | 20.4–31.4 | 12–17.6 |
| Coastal bermudagrass [b] | 25 | 37.5 | 6.4 |
| Leaves [b] | 15–20 | 80–85 | 0 |
| Wheat straw [b] | 30 | 50 | 15 |
| Cottonseed hairs [b] | 80–95 | 5–20 | 0 |
| Nutshells [b] | 25–30 | 25–30 | 30–40 |
| Corn stover [c] | 37.5 | 22.4 | 17.6 |
| Corn fibres [c] | 14.28 | 16.8 | 8.4 |
| Pinewood [c] | 46.4 | 8.8 | 29.4 |
| Office paper [c] | 68.6 | 12.4 | 11.3 |
| Sugarcane bagasse [c] | 40–50 | 25–35 | 17–20 |

[a] Klímek et al. (2017) [58], [b] Kumar et al. (2009) [56] and [c] Mosier et al. (2005) [55].

Prominent factors that determine the contents of cellulose, hemicellulose, and lignin are the growth region and conditions of the plants [59]. Agricultural residues, agro-wastes, forest wastes, industrial wastes, municipal solid wastes, and forest biomass, to mention but a few, are the commonly used lignocellulosic sources (Figure 4) [60–64].

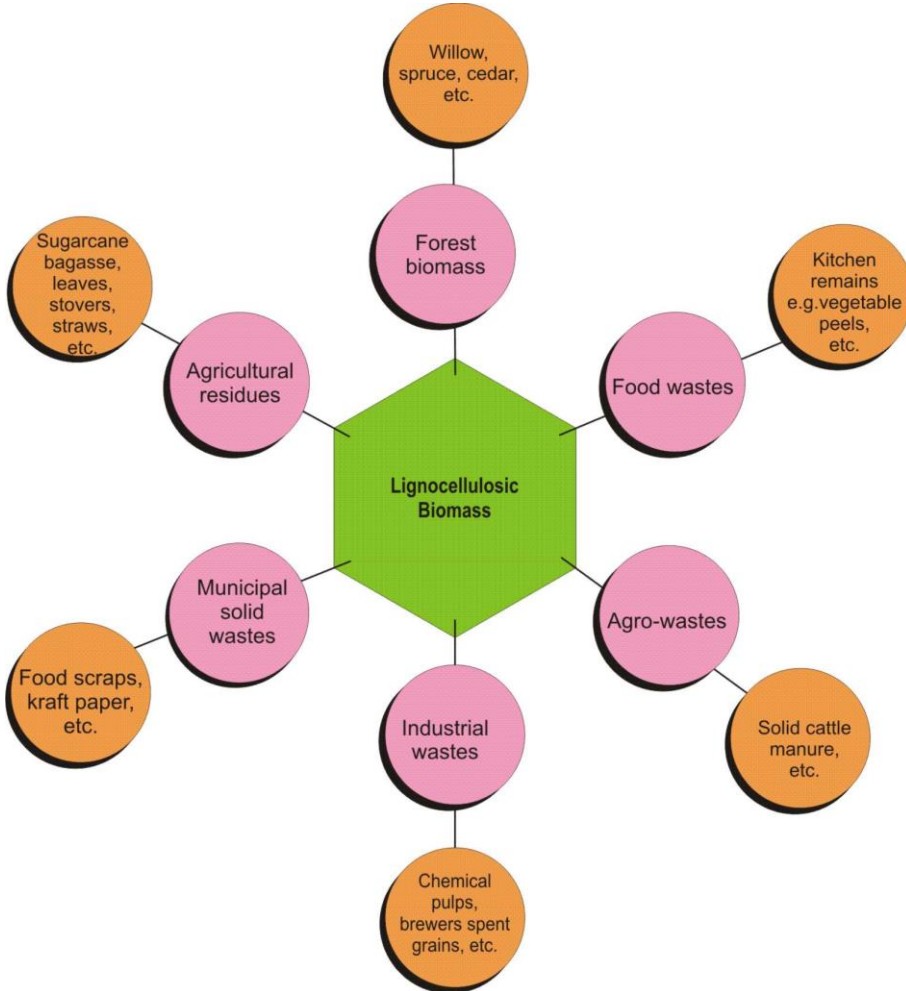

**Figure 4.** Block diagram of different lignocellulosic biomass sources (Data from Sharma et al. (2023) [65], Adewuyi (2022) [66], and Kumar et al. (2009) [56]).

Sugarcane bagasse and brewer spent grains are promising lignocellulose mass recently used as feedstock for lactic acid production. Sugarcane bagasse is a fibrous residue obtained

after the juice has been extracted from the sugarcane stalk [67,68]. The brewer's spent grains are a brewing by-product obtained during beer brewing [69]. Converting lignocellulosic biomass into lactic acid involves four process steps: pretreatment, hydrolysis, fermentation, and product separation or purification using a suitable microorganism(s) operating at a selected operational mode.

### 4. Pretreatment Processes of Lignocellulosic Materials

Due to the complexity of the lignocellulosic materials, pretreatment is a key technology used in bio-based industries [70]. The pretreatment of lignocellulosic materials allows the removal of lignin and hemicellulose, reduces the crystallinity of cellulose, and increases the porosity of lignocellulosic materials [56]. In the pretreatment step, biomass macroscopic and microscopic, and chemical compositions are disrupted to achieve more rapid hydrolysis of carbohydrate polymer that produces greater yields of monomeric sugars [55]. Different methods employed in pretreating lignocellulosic materials, as presented in Figure 5, include physical (grinding or milling, pyrolysis, etc.), physicochemical (steam explosion, ammonia fiber explosion (AFEX), carbon dioxide, etc.), chemical (e.g., dilute acid hydrolysis, alkaline hydrolysis, organosolv process, oxidative delignification, ozonolysis, etc.), biological (using ligninolytic enzymes of certain microorganisms) pretreatments [51,52,71,72].

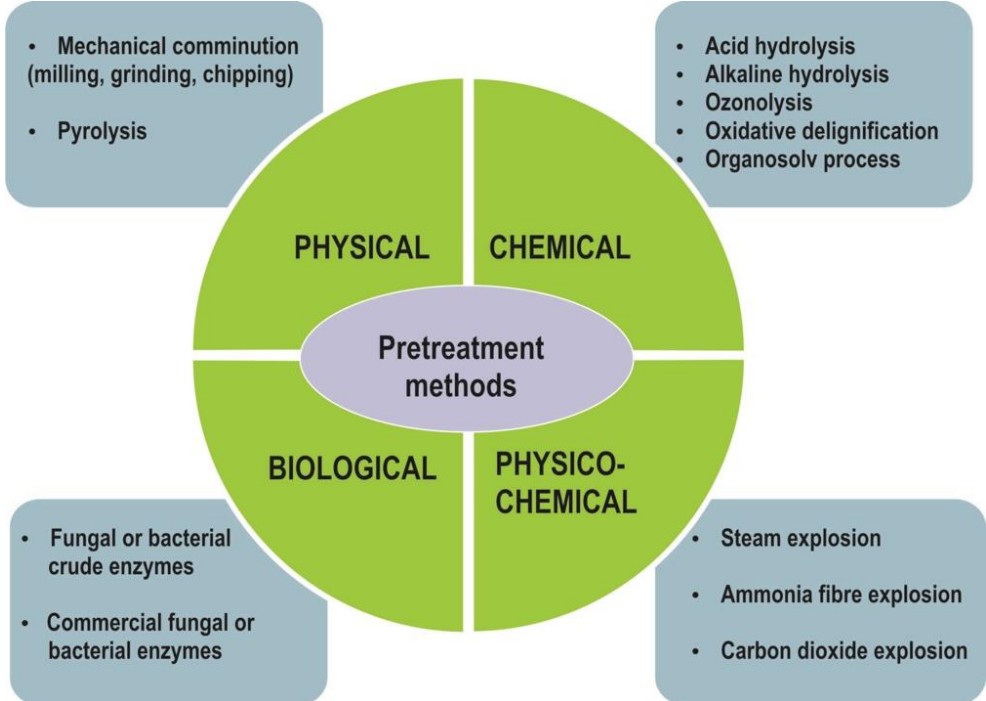

**Figure 5.** Different pretreatment methods of lignocellulosic materials for saccharification before the fermentation process (Adapted from Rawoof et al. (2020) [29]).

The pretreatment process of lignocellulosic materials is critical and must be sufficiently effective for the resultant residue to be easily saccharified by hydrolytic enzymes and should not be overprocessed to avoid the production of toxic compounds that can inhibit microorganisms' metabolism and growth [33]. Effective pretreatment methods should be cost-effective; the pretreatment methods should improve sugar formation, produce less reduction and avoid carbohydrate loss and formation of inhibitory by-products such as organic acids, phenolic compounds, and aldehydes [33,56,73]. Different approaches used in lignocellulosic biomass pretreatment and their advantages and disadvantages are listed in Table 2.

**Table 2.** A summary of various lignocellulosic biomass pretreatment processes and quantitative [56,73–75].

| Pretreatment Method | Advantages | Disadvantages |
|---|---|---|
| Mechanical comminution | (i). Reduces cellulose crystallinity | (i). High power consumption |
| Pyrolysis | (i). Gas and liquid production | (i). High temperature required<br>(ii). Production of ash |
| Steam explosion | (i). Cost-effective<br>(ii). Hemicellulose solubilization and lignin transformation<br>(iii). High yield of glucose and hemicellulose in a two-step process | (i). Incomplete lignin degradation<br>(ii). Partial degradation of the xylan fraction<br>(ii). Not efficient for biomass with high lignin content<br>(iv). Toxic compounds such as acetic acid and a small amount of furan aldehydes generation |
| Ammonia fibre explosion | (i). Removes some lignin and hemicellulose<br>(ii). Low formation of inhibitors<br>(iii). Increases the accessible surface area; thus, cellulose becomes more accessible<br>(iv). Does not need a small particle size for efficacy | (i). Not effective for high lignin content biomass<br>(ii). Recycling ammonia is needed<br>(iii). Alters lignin structure<br>(iv). High cost of ammonia |
| Carbon dioxide explosion | (i). Cost-effective<br>(ii). Increases the accessible surface area<br>(iii). No inhibitory compounds generated | (i). No modification of lignin or hemicellulose can be made |
| Acid hydrolysis | (i). High glucose yield<br>(ii). Solubilizes hemicellulose to xylose and other sugars<br>(iii). Alters lignin structure | (i). High cost<br>(ii). High cost of corrosive-resistant equipment<br>(iii). Inhibitors such as aliphatic carboxylic acids (acetic acid, formic acid, levulinic acid) are generated |
| Alkaline hydrolysis | (i). Efficient removal of lignin and hemicellulose<br>(ii). Increases the accessible surface area<br>(iii). Low inhibitor generation | (i). Long residence time required<br>(ii). Irrecoverable salts were incorporated into biomass<br>(iii). High cost of alkaline catalyst<br>(iv). Alteration of lignin structure |
| Ozonolysis | (i). Reduction of lignin content<br>(ii). No toxic compounds generation | (i). A large amount of ozone is required, thus making the process expensive |
| Oxidative delignification | (i). Degrades lignin<br>(ii). Low inhibitor generation | (i). Not all oxidizing agents are effective for delignification |
| Organoslv process | (i). Hydrolyzes lignin and hemicellulose | (i). Requires solvent to be drained from the reactor, and it must be evaporated, condensed, and recycled<br>(ii). High cost |
| Biological pretreatment | (i). Degrades lignin and hemicellulose<br>(ii). Requires low energy | (i). Slow hydrolysis process |

## 4.1. Physical Pretreatment Methods

Mechanical comminution and pyrolysis are among the physical pretreatment methods. In mechanical comminution, lignocellulosic materials are combined by grinding, milling, and chipping to reduce cellulose crystallinity. This results in a 10–30 mm material size after chipping and 0.2–2 mm after grinding or milling [76]. Comminution is an energy-intensive operation, and the power used in mechanical comminution depends on biomass characteristics and the final particle size [77]. In pyrolysis, a temperature greater than 300 °C is applied to lignocellulosic material, and cellulose rapidly decomposes to form gaseous products and residual char.

### 4.2. Physicochemical Pretreatment Methods

Steam explosion, ammonia fiber explosion (AFEX), and carbon dioxide ($CO_2$) explosion are physicochemical pretreatments usually used in treating lignocellulosic materials [56]. Different physical factors affect the effectiveness of steam explosion pretreatment; these include temperature, residence time, material size, and moisture content [74,78].

In a steam explosion where water acts as an acid at high temperature, lignocellulosic material undergoes an explosive decompression when high-pressure saturated steam is initially applied to the material, followed by a pressure reduction (at atmospheric pressure). During the steam explosion, the temperature is usually initiated at a 160–260 °C range temperature, corresponding to 0.69–4.83 MPa pressure. At a high temperature (e.g., 90 °C) and pressure, lignocellulose materials are exposed to liquid ammonia for 30 min, followed by a swift pressure reduction [56]. The high temperature causes hemicellulose to degrade and transform lignin, increasing the possible cellulose hydrolysis [76]. However, studies have shown that adding dilute acid such as sulphuric acid in a steam explosion can improve hydrolysis, thereby leading to complete hemicellulose removal [76].

The ammonia fiber explosion (AFEX) process is similar to the steam explosion process; in AFEX pretreatment, the ammonia dosage is usually 1–2 kg of ammonia per dry lignocellulose material (kg) at 90 °C [76]. Though in AFEX pretreatment, hemicellulose is not significantly solubilized compared to the acid-catalyzed steam explosion, the pretreatment has been used to pretreat bagasse, alfalfa, wheat chaff, wheat straw, coastal Bermuda grass, switchgrass, to mention but a few [79–81].

Pretreatment of lignocellulose materials using carbon dioxide ($CO_2$) explosion aimed to improve lignocellulosic pretreatment techniques and reduce the cost expense of steam explosion and ammonia explosion [56]. Zheng et al. (1998) [82] showed that $CO_2$ explosion is more cost-effective, and no inhibitory compounds were formed, unlike in ammonia explosion pretreatment. In a $CO_2$ explosion, supercritical fluid in a gaseous form is compressed to a liquid density at the temperature above its critical point; upon the explosive release of $CO_2$ pressure, the disrupted cellulosic structure increases the substrate's accessible surface area for hydrolysis [56,83].

### 4.3. Chemical Pretreatment Methods

Chemical pretreatment has been extensively used in removing lignin surrounding cellulose. Various chemical pretreatment methods employed in treating lignocellulosic materials include acid hydrolysis, alkaline hydrolysis, ozonolysis, oxidative delignification, and organosolv process.

Acid hydrolysis, using concentrated acids to pretreat lignocellulosic materials, is effective in cellulose hydrolysis. However, concentrated acids are corrosive, hazardous, and toxic; pretreatment of lignocellulosic materials using concentrated acids requires a corrosion-resistant reactor or consistent reactor maintenance to minimize reactor corrosion [56,76]. Dilute acid hydrolysis has been successfully used in pretreating lignocellulosic materials. The two types of dilute acid pretreatment processes include the low temperature (<160 °C) batch process for high solids loading (10–40% [weight of substrate/weight of reaction mixture]) and the high temperature (>160 °C) continuous-flow process for low solids loading [84–86]. Using dilute acid in pretreating lignocellulosic materials at a high-temperature favors cellulose hydrolysis. However, at a moderate temperature, direct saccharification is affected by low yields due to sugar decomposition [86,87]. Though dilute acid pretreatment improves cellulose hydrolysis, neutralizing pH after the pretreatment is required for the downstream enzymatic hydrolysis. The mechanism of alkaline hydrolysis in pretreating lignocellulosic materials could be linked to the saponification of intermolecular ester bonds linking xylan, other hemicelluloses, and lignin and the removal of the crosslinks results in the increased porosity of the lignocellulosic material.

Using NaOH in pretreating lignocellulosic causes swelling, resulting in an internal surface area increase, reduction in the degree of polymerization, reduction in crystallinity, structural linkages between lignin and carbohydrates separation, and the lignin struc-

ture disruption [88,89]. Lignocellulosic material response to diluting NaOH may differ depending on the lignin contents [90]. For instance, Feist et al. (1970) [91] reported that the hydrolysis of hardwoods and other lignocellulosic materials with low lignin contents significantly increased when treated with NaOH, while hydrolysis of softwoods with high lignin contents only increased slightly with NaOH pretreatments. In addition, Iyer et al. (1996) [92] reported the efficiency of delignification was 60–80% for the mixture of corn hobs and stover and 65–85% for switchgrass after ammonia pretreatment to remove lignin.

In ozonolysis, ozone is used to degrade mostly lignin, while hemicellulose is slightly attacked, but cellulose is hardly affected. Several studies have reported an increase in enzymatic hydrolysis rate and yield due to the decrease in lignin contents of some lignocellulosic materials, such as wheat straw and popular sawdust, after ozonolysis pretreatment [93]. Additionally, Hermansyah et al. (2021) [94] applied ozone for 90 min at pH 3 in the delignification of sugarcane bagasse and observed the production of cellulose, hemicellulose, and lignin estimated at 59%, 22%, and 6%; thus, ozonolysis reduced lignin up to 217%. The advantages of ozonolysis include the effective removal of lignin, no production of toxic residues, and the reaction being carried out at room temperature and pressure [93]. However, a large amount of ozone is required, thus making it expensive.

In oxidative delignification, oxidizing agents such as hydrogen peroxide ($H_2O_2$), sodium chlorite ($NaClO_2$), sodium hypochlorite (NaOCl), potassium permanganate ($KMnO_4$), potassium peroxydisulphate ($K_2S_2O_8$), nitrogen oxide (NO), chlorine dioxide ($ClO_2$), or sulfur dioxide ($SO_2$) is used in pretreating lignocellulosic materials [89,95–100]. During oxidative delignification, many free radicals released from the oxidizing agents cause oxidative fragmentation and lignin removal from the lignocellulosic materials [101]. In addition, cellulose is structurally modified when these oxidizing agents penetrate cellulose and then oxidize it [89]. Different oxidizing agents used in the delignification of lignocellulosic materials have been reported. Park et al. (2015) [100] reported that sodium chlorite delignification is the most effective process in removing lignin from hardwoods, while hydrogen peroxide delignification was less effective than sodium chlorite delignification and oxygen delignification. Costa et al. (2018) [96] reported similar observations when banana rachis was subjected to delignification; treatments with active chlorine, e.g., sodium chlorite and hypochlorous acid, were more effective than hydrogen peroxide delignification. The latter left great amounts of residual lignin in the samples. However, pretreatment of cane bagasse with alkali hydrogen peroxide resulted in most hemicellulose and 50% lignin solubilization leading to 95% efficiency of glucose production during enzymatic hydrolysis [102]. Mutrakulcharoen et al. (2021) [95] have shown that lignocellulosic material (rice straw) treated with potassium permanganate enhanced the reducing sugar yield.

Organosolv processing breaks the internal lignin and cellulose bonds by an organic or aqueous organic solvent mixture with inorganic acid catalysts (e.g., $H_2SO_4$ or HCl). The summary of reactions associated with organosolv pretreatment includes hydrolysis of ether and 4-*O*-methylglucuronic acid ester bonds to the α-carbons of the lignin units and hydrolysis of the glycosidic bonds in the hemicellulose [76,103]. The organic solvents used in the organosolv process include methanol, acetone, and ethanol, to mention but a few, while oxalic, acetic acid, formic acid, and salicylic acid are among the organic acids that can be used in the process [103].

### 4.4. Biological Pretreatments

Physical, physicochemical, and chemical pretreatments of lignocellulosic materials are highly effective but expensive due to the high energy consumption; chemicals can produce streams with high-contaminant potential and generate by-products that can inhibit downstream processes [104]. In contrast, biological pretreatments are mediated by microorganisms and their enzymes under mild conditions, such as ambient temperature and pressure, that require low energy and chemicals with reduced inhibitor production [105].

Ligninolytic enzymes, a group of oxidoreductases, degrade lignin, thus improving biomass degradation. Ligninolytic enzymes include laccases (found in plants, fungi, and

bacteria) and heme peroxidases [lignin peroxidases (LiPs), manganese peroxidases (MnPs), versatile peroxidases (VPs) and dye-decolourizing peroxidases (DyPs)] with high redox potential that directly oxidize lignin polymers [106]. In addition, in biological pretreatments, the extracted ligninolytic enzymes or microorganisms that secret ligninolytic extracellular oxidative enzymes are used in degrading lignin and disrupting its structure, thus allowing access to holocellulose by hydrolytic enzymes in the release of sugars [107]. Some fungi or bacteria are involved in the microbial depolymerization of lignocellulosic materials, but wood decay fungi are the most common microorganisms responsible for this process [89]. These microorganisms are classified into white rots (e.g., *Fomes fomentarius*, *Phellinus igniarius*, *Ganoderma appalanatum, and Pleurotus ostreatus*) that attack lignin and cellulose, red rots (e.g., *Fomitopsis annosa*) that attack lignin and cellulose and brown rots (e.g., *Piptoporus betulinus*, *Gloeophyllum sepiarium*, *Fomitopsis pinicola*, *Trametes quecina*, *Laetiporus sulphureus*) that attack the cellulose [89,104,106]. Of these, the most effective lignin-degrading microorganisms are the white rots, Basidiomycetes [89,108,109]. However, biological pretreatments need improvements to be a suitable alternative to the other pretreatments as their processes are slow [104].

## 5. Hydrolysis Processes of Treated Lignocellulosic Biomass

A hydrolysis step is required for fermentative lactic acid production. In hydrolysis, cellulose, a linear polymer of numerous D-glucose linked by β-(1→4)-glycosidic bonds and hemicelluloses, branched heteropolymers of hexoses, pentoses and uronic acids linked by structural linkages are depolymerized into fermentable sugars [55,107]. The crystalline structure of cellulose makes it more challenging to hydrolyze, unlike hemicellulose, which is amorphous [110]. Both enzymatic and acid hydrolysis of cellulose and hemicellulose have been reported, but cellulose hydrolysis using concentrated mineral acids is neither safe nor suitable for the environment due to the formation of the abundance of salts during neutralization [110]. Enzymatic hydrolysis of pretreated lignocellulosic materials is the most promising means of producing fermentable sugars [33].

Groups of enzymes used in converting cellulose and hemicellulose to fermentable sugars are cellulases and hemicellulases [60,111]. The mixture of cellulase and hemicellulase increases the hemicellulose hydrolysis and thus increases cellulase access resulting in a decrease in hydrolysis time and process cost [33,112]. The rate of enzymatic hydrolysis of cellulose, a homopolymer, depends on the cellulose polymerization degree. Three enzymatic activities that hydrolyze cellulose include random hydrolysis of internal β-(1→4)-glycosidic bonds in the cellulose chain by endoglucanases, cleavage of cellobiose units from the end of the chain by cellobiohydrolases and conversion of cellobiose to glucose by cellobiohydrolases [107]. This mixture of cellulases (endoglucanases, cellobiohydrolases, and ß-glucosidase) acts synergistically to efficiently hydrolyze and saccharify cellulose and mitigate product inhibition [33,107].

Hemicelluloses are heteropolymers grouped according to the sugar residue in the highest quantity in the polysaccharide chain and include xylan, galactan, arabinan, and mannan polymers [113]. Hemicellulose is easily accessible because it does not form tight crystalline structures. It has a more varied structure and composition than cellulose, and as a result, more enzymes are required in its hydrolysis than in cellulose hydrolysis [107]. The core enzymes, endoxylanases, cut the main polysaccharide chain of xylan into shorter oligosaccharides, and β-xylosidase separates short xylooligosaccharides into xylose in enzymatic hydrolysis of xylan to monomers [114]. In addition, the lateral groups such as arabinose, acetyl, glucose, and galactose linked to the main polysaccharide chain are cleaved by α-L-arabinofuranosidases, α glucuronidase, acetyl mannan esterase, acetyl xylan esterase, feruloyl esterase, ρ-coumaric acid esterase and ferulic acid esterase [107,113]. In cellulose hydrolysis, endoglucanases or cellobiohydrolases hydrolyze cellulose to cellobiose, and cellobiose is hydrolyzed by β-glucosidase into glucose. In hemicellulose hydrolysis, endoxylanases hydrolyze hemicellulose to form oligosaccharides, and β-xylosidase hydrolyzes oligosaccharides to form xylose whereas endomannanases hydrolyze hemicellulose to

form oligosaccharides, and mannose is formed through the hydrolysis of oligosaccharides by β-mannosidase [107].

Enzymatic hydrolysis of lignocellulosic materials is the most notable technology for biomass saccharification. Different hydrolytic enzymes produced by microorganisms and commercially available options are listed in Table 3.

**Table 3.** Enzymatic hydrolysis of pretreated lignocellulosic materials using selected commercial enzymes and crude enzymes obtained directly from microorganisms.

| Sources | Pretreatment | Enzyme | Performance | References |
|---|---|---|---|---|
| Sugarcane bagasse | Sulphite-NaOH treatment at 140 °C for 30 min | * Cellic CTec2 immobilized on GO-MNP[a] | Cellulose conversion into 74% of glucose. Xylan conversion into 74% of glucose | [115] |
| Sugarcane bagasse | Chlorite-acetic acid treatment at 70 °C for 4 h | * Cellic CTec2 immobilized on GO-MNP [a] | Cellulose conversion into 54% of glucose. Xylan conversion into 36% of glucose | [115] |
| Sugarcane bagasse | Dilute hydrochloric acid at 96.8 °C for 375 min | # Endocellulase in DIOMNP and β-glycosidase in GLA [a] | Conversion of approximately 39.06% of cellulose into 94.54% of glucose | [116] |
| Wheat straw | Soaked in H$_2$SO$_4$ (0.2%) at 190 °C for 10 min | Celluclast [a] and Novozyme 188 [a] | Conversion to 60% glucose | [117] |
| Spruce | Impregnated with SO$_2$ (2.5%) at 210 °C for 5 min | Celluclast [a] and Novozyme 188 [a] | Conversion to approximately 29% glucose | [117] |
| Bagasse | Impregnated with SO$_2$ (2.5%) at 200 °C for 5 min | | Conversion to 50% glucose | |
| Spruce | Impregnated with SO$_2$ (2.5%) at 210 °C for 5 min | *Trichoderma atroviride* crude enzyme supernatant [b] | Conversion to approximately 29% glucose | |
| Wheat straw | Soaked in H$_2$SO$_4$ (0.2%) at 190 °C for 10 min | | Conversion to 64% glucose | |
| Bagasse | Impregnated with SO$_2$ (2.5%) at 200 °C for 5 min | | Conversion to 52% glucose | |
| Switchgrass | (i) SG-AC (ii) SG-HA | Cellulase NS50013 [a] + β-glycosidase NS50010 [a] | Conversion to 96–98% glucose | [118] |
| Rice straw | Acidifies steam explosion (SWAN) | (i) Cellulase [a] (ii) Cellulase [a] + Novozyme 188 [a] | (i) Cellulose conversion into 34.2% glucose (ii) Cellulose conversion into 45.7% glucose | [119] |
| | Dilute sulphuric acid (0.8%) at 160 °C for 10 min | (i) Cellulase [a] (ii) Cellulase [a] + Novozyme 188 [a] | (i) Cellulose conversion into 35.4% glucose (ii) Cellulose conversion into 59.0% glucose | |
| | AFEX (reactor and sample temperatures: 74 °C and 70 °C) at 350 psi for 20 min | (i) Cellulase [a] (ii) Cellulase [a] + Novozyme 188 [a] | Cellulose conversion into 28.5% glucose (ii) Cellulose conversion into 32.0% glucose | |

GO-MNP: Magnetic graphene oxide particles; *: 20 FPU.g$^{-1}$; #: 1.5 U; [a]: Commercial enzymes; [b]: crude enzyme directly from microorganism; SG-AC: Dilute acid and SG-HA: Hypochlorite-alkaline methods.

## 6. Fermentation of Sugars

The fermentation process involves the biological degradation of a substrate, e.g., glucose, by a group of microorganisms into metabolites such as lactic acid, ethanol, etc. [120].

The metabolites produced as the fermentation end-products depend on the microorganisms used for the fermentation process. Microorganisms used in fermentation are divided into two groups: bacteria and fungi [120]. Different modes used in the fermentation processes and their advantages and disadvantages have been reported [30].

### 6.1. Lactic Acid-Producing Bacteria (LAB) and Their Fermentative Pathway

Bacteria involved in lactic acid production include lactic acid-producing bacteria (LAB), *Bacillus* sp., and *Corynebacterium glutamicum* [121–123]. Lactic acid-producing bacteria (LAB) cannot synthesize ATP through respiration, but their main end-product of sugar fermentation is lactic acid [120]. Most LAB are facultative anaerobes, non-motile and non-spore-forming; their optimal growth temperature varies from 20 to 45 °C, and the optimal pH ranges from 5.5 to 6.5 (depending on the species) [29]. LAB require complex nutrients such as amino acids, vitamins, minerals, and carbohydrates [124]. Lactic acid-producing bacteria are classified into homofermentative and heterofermentative according to the fermentation end-product [125]. Homofermentative LAB convert glucose to lactic acid (primary by-product), while heterofermentative LAB, obligatory or facultative, converts glucose to lactic acid, ethanol or acetate, and carbon dioxide [30]. In homolactic fermentation (Figure 6A), homofermentative LAB such as *Lactobacillus acidophilus*, *Lactobacillus amilophylus*, *Lactobacillus helveticus*, *Lactobacillus bulgaricus*, *Lactobacillus salivarius*, convert glucose via the Embden-Meyerhof-Parnas pathway (glycolysis), resulting in lactic acid as the end-product of glucose metabolism [3,30]. However, some homofermentative LAB can produce formic acid or mixed acid fermentation (Figure 6B) by pyruvate-formate lyase when there is a stress condition such as carbon source limitation or increased pH or decreased temperature [30,126].

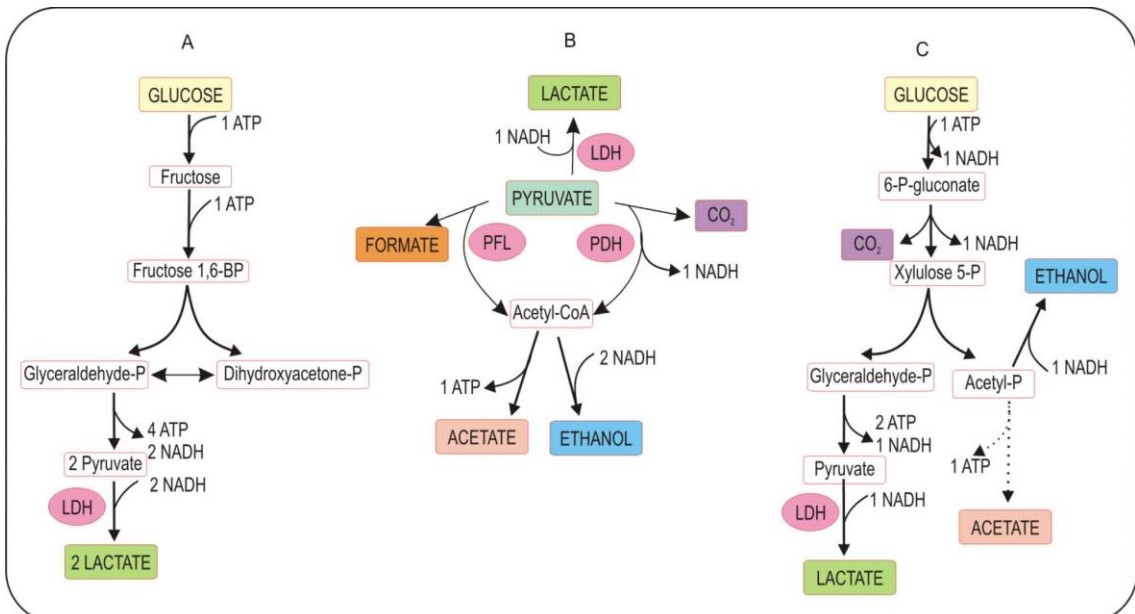

**Figure 6.** Catabolic pathways of glucose fermentation by lactic acid-producing bacteria (LAB). (**A**): homofermentation in *Lactococcus lactis* or *Lactobacillus acidophilus*, (**B**): mixed acid fermentation (*Lactococcus lactis*), and (**C**): heterofermentation in *Lactobacillus casei*. P: phosphate, BP: bisphosphate, PDH: pyruvate dehydrogenase, PFL: pyruvate formate lyase, and LDH: lactate dehydrogenase (Data from Johanson et al. (2020) [126] and Wang et al. (2021) [127].

Theoretically, during homolactic fermentation (Figure 6A), the molar yield should be 2 moles of lactic acid per mole of consumed glucose with the yield of 1 g of product per g

of the substrate (Equation (1)). Still, studies have shown that the experimental lactic acid yield is usually lower [3].

$$\text{Glucose} \rightarrow 2 \text{ lactic acid} + 2 \text{ ATP} \tag{1}$$

Most homofermentative LAB are available for commercial lactic acid production [3] and are believed to mostly belong to the genus *Lactobacillus* [30]. In heterofermentation (Figure 6C), facultative heterofermentative LAB such as *Lactobacillus alimentarius*, *Lactobacillus casei*, *Lactobacillus plantarum*, *Lactobacillus lactis*, *Lactobacillus pentosus* and *Lactobacillus xylosus*, ferment sugar by the 6-phosphogluconate and phosphoketolase pathways. Obligatory heterofermentative LAB, such as *Lactobacillus brevis*, *Lactobacillus fermentum,* and *Lactobacillus reuteri*, ferment sugar by either the 6-phosphogluconate pathway or phosphoketolase pathways [3].

In general, lactic acid bacteria are high acid-tolerant; they are of great commercial importance as they can be engineered for elective production of L- or D- lactic acid [128]. However, some drawbacks of using LAB for commercial lactic acid production are the high complex nutrient requirements, which may increase the production cost, and the fermentation temperature, which could pose contamination risks [46].

### 6.2. Lactic Acid-Producing Fungi

Other microorganisms that can produce lactic acid are filamentous fungi, e.g., *Rhizopus oryzae* and *Rhizopus arrhizus*, which produce L-lactic acid as the main fermentation product. These filamentous fungi have similar metabolic pathways. A model of glucose metabolic pathways in *Rhizopus oryzae* revealed two separately regulated pools of pyruvate in the organisms. These two regulated pyruvate pools consist of a cytosolic pyruvate pool channeled into ethanol, lactate, oxaloacetate, malate, and fumarate synthesis, and the second pool of pyruvate (in the mitochondrion) channeled into a tricarboxylic acid cycle [129].

Lactic acid production using several renewable resources by *Rhizopus* strains has been reported. Bai et al. [130] reported the production of over 77.2 g·L$^{-1}$ L-lactic acid from corncob xylose by *Rhizopus oryzae* strain HZS6. In addition, Miura et al. [131] reported enhanced L-lactic acid production from an untreated raw corncob using a mixed culture of *Rhizopus* sp. MK-96-1196 (L-lactic acid producer) and *Acremonium thermophilus* ATCC 24622 (a cellulose producer). Waste office paper, wheat straw and xylose, chicken feather protein hydrolysate, sugar beet, and molasses, to mention but a few, were used in lactic acid production by *Rhizopus oryzae* [132–134]. Zhang et al. [135] reported 88 g·L$^{-1}$ of lactic acid production using waste potato starch by acid-adapted preculture *Rhizopus arrhizus* in a bubble column reactor. Huang et al. [136] reported lactic acid production with a yield of 0.85–0.92 g·g$^{-1}$ using potato starch wastewater by *Rhizopus arrhizus* and *Rhizopus oryzae*. Nonetheless, some strains of *Rhizopus oryzae* produce a small amount of fumaric acid and ethanol during fermentation [134,137].

### 7. Genetically Modified Microorganisms

Genetic modification approaches have been used to improve lactic acid-producing microorganisms' thermotolerance, osmotolerance, and resistance to lignocellulosic hydrolyzate inhibitors. Yeasts, such as *Saccharomyces cerevisiae*, do not produce lactic acid naturally, but the metabolic engineering of yeasts allows lactic acid production during fermentation through the exchange of ethanol with lactic acid. *Saccharomyces cerevisiae*, *Candida sonorensis*, *Candida boidinii*, *Candida utilis,* and some *Kluyveromyces* sp. were reported to produce lactic acid after genetic modifications [138]. Tolerance of yeast to low pH leads to a reduction in the need for neutralizing agents, hence, reducing the downstream processing cost. However, the disadvantage of using wild-type yeast is reduced lactic acid production; yeasts are engineered to overcome this drawback [46]. Ikushima et al. [139] reported the production of L-lactic acid (103.3 g·L$^{-1}$) from pulping waste liquors by the engineered *Candida utilis*, while Pecota et al. [140] reported the production of lactic acid (over 24 g·L$^{-1}$) using a genetically engineered *Kluyveromyces marxianus*. Lactic acid (85.9 g·L$^{-1}$)

was produced by genetically modified *Candida boidinii* [141], and modification of *Candida sonorensis* resulted in the accumulation of 92 g·L$^{-1}$ lactic acid without ethanol production [142]. Engineered *Saccharomyces cerevisiae* BK01 was reported to produce 119 g·L$^{-1}$ of lactic acid without using pH neutralizers [143]. Genetic engineering can improve lactic acid production by decreasing or deleting pyruvate decarboxylase (PDC) activity in *Saccharomyces cerevisiae* [144]. The relevant metabolic pathway of lactic acid production in yeast is presented in Figure 7.

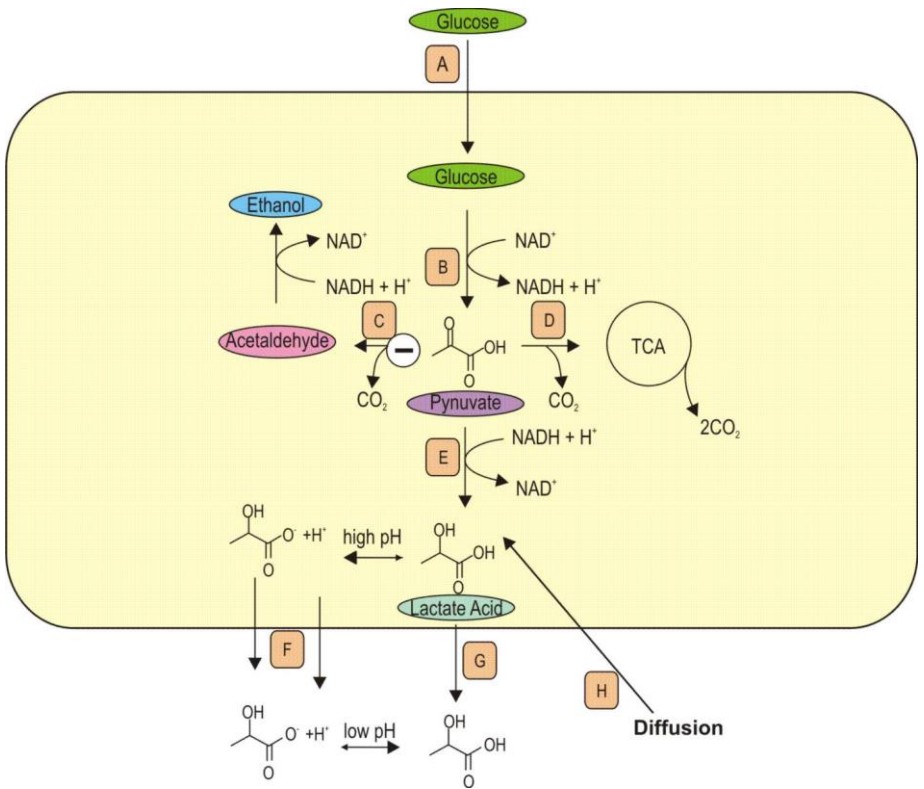

**Figure 7.** Schematic diagram of the metabolic pathway of lactic acid production in genetically modified *Saccharomyces cerevisiae*. A: Transport of glucose via hexose transporters; B: The release of 1ATP per molecule of pyruvate formed (glycolysis); C: Non-oxidative decarboxylation of pyruvate to acetaldehyde and carbon dioxide by pyruvate decarboxylase (PDC); −: Decrease or deletion of PDC activity; D: Oxidative decarboxylation of pyruvate catalyzed by pyruvate dehydrogenase via the tri-carboxylic acid (TCA) cycle; E: Lactic acid production from pyruvate catalyzed by lactate dehydrogenase as a result of introducing heterologous activity by genetic engineering; F: Lactate/H$^+$ symport (for exporting lactic acid); G: Lactic acid export from the cell and H: Lactic acid protonation and diffusion into the cell (when the extracellular lactic acid is low) (Adapted from Zhu et al. (2022) [145]).

## 8. Different Modes Used in Fermentative Lactic Acid Production

The selection of fermentation modes varies with respect to the substrate's nature, fermentation broth's viscosity, and microorganisms used and their growth [23,33]. Different fermentation modes used in lactic acid production include batch, fed-batch, repeated, and continuous [30]. These fermentation modes have several advantages and disadvantages. The advantages of batch fermentation include ease of operation, high lactic acid concentration and yield, and minimal contamination risks. However, batch fermentation exhibits low productivity and substrate or product inhibition [146] unlike in fed-batch, where there is limited by-product accumulation, high product concentration, and no substrate inhibition. The optimal design of the fed-batch fermentation and end-product inhibition are fed-batch disadvantages [147]. Although repeated batch fermentation requires special devices (e.g., special connection lines for cell concentration), this fermentation mode saves time and labor, and the mode gives a high growth rate [147]. High lactic acid productivity and

microorganism growth control rate are the advantages of continuous fermentation [146]. The drawbacks of continuous fermentation include high risks of contamination, incomplete usage of carbon sources, and high cost of equipment [147].

### 8.1. Batch Fermentation Mode

In the batch fermentation mode, all the required nutrition is added before the fermentation process starts. In most cases, acid or alkaline control is added to the system to maintain a constant pH value [33]. Batch fermentation is a closed system; as such, the risk of contamination is reduced, and the production of high lactic acid concentrations has been reported [30,33]. Abdel-Rahman et al. [148] reported the production of 119 g·L$^{-1}$ L-lactate during batch fermentation of cellobiose using LAB, and 80 g·L$^{-1}$ D-lactate was produced using hydrolyzed cane sugar in the fermentation medium [149]. The drawbacks of batch fermentation mode include nutrient depletion that limits microbial cell concentration and low productivity due to substrate or product inhibition [23,33]. However, the cell mass, lactic acid production, and productivity were increased when nutrients were added to the broth of *Enterococcus mundtii* QU 25 during batch fermentation [148]. Different methods, such as separate hydrolysis and fermentation (SHF), solid-state fermentation (SSF), and simultaneous saccharification and fermentation (SmSF) used in batch setups, will be discussed later.

### 8.2. Fed-Batch Fermentation Mode

The fed-batch fermentation mode is a modified version of batch fermentation. It contains the same required components, such as raw materials (carbon source), nitrogen source, and other essential nutrients, as in batch fermentation. Nonetheless, during the fed-batch fermentation process, one or more of the required nutrients are fed aseptically at regular intervals without removing any fermentation broth [150,151]. In fed-batch fermentation, the amount of limiting nutrients such as carbon and nitrogen added sequentially at regular intervals determines the reaction rates. In addition, adding limiting nutrients enhances microbial growth, resulting in higher yield. Fed-batch fermentation is more advantageous than batch fermentation. Liu et al. [152] reported that fed-batch culture carried out by keeping the glucose concentration at 30 g·L$^{-1}$ resulted in over 140 g·L$^{-1}$ L-lactic acid production with a product yield of 83%. However, the batch culture with 200 g·L$^{-1}$ initial glucose concentration produced 121 g·L$^{-1}$ L-lactic acid with a low product yield. In addition, Paulova et al. [151] reported the maximum final L-lactic acid concentration (116.5 g·L$^{-1}$) using pulse-fed fed-batch fermentation. Besides an increase in yield and productivity in fed-batch, there is a long-term synthesis of a product, low substrate concentration, and reduction in substrate inhibition due to the regular supply of nutrients to the fermentation culture [33].

### 8.3. Repeated Batch Fermentation Mode

The repeated batch fermentation mode is similar to the batch mode, wherein all the required nutrients are added at the start. However, in repeated batch fermentation, over time, the depleted nutrients are replaced with an equal volume of fresh medium containing the same nutrients and culture broth as the initial fermentation broth (microbial cells recycling) for the further fermentation process [20,153]. Reports have shown that in repeated batch fermentation, lactic acid productivity increased. Reddy et al. [154] reported a significant increase in lactic acid productivity from 3.20 to 6.37 g·L$^{-1}$·h$^{-1}$ when a total of 10 repeated-batch fermentations were carried out using 100 g·L$^{-1}$ hydrol, 150 g·L$^{-1}$ soya bean curd residue hydrolyzate, and 20 g·L$^{-1}$ malt hydrolyzate as the main nutrients. Additionally, Yin et al. [155] observed average L-lactic acid productivity of 2.02 g·L$^{-1}$·h$^{-1}$, which was 1.9-fold higher than that from batch culture during the first six cycles of repeated batch culture. Furthermore, Abdel-Rahman et al. [156] reported a great improvement in lactic acid productivity by new alkaliphilic bacterium using repeated batch fermentation. For instance, in batch fermentation mode using 20 g·L$^{-1}$ glucose, 19.6 g·L$^{-1}$ of lactic acid was produced with 2.18 g·L$^{-1}$·h$^{-1}$

volumetric productivity. While in repeated batch mode, there was an increase in lactic acid productivity up to 39.9 g·L$^{-1}$·h$^{-1}$ using 40 g·L$^{-1}$ glucose. Furthermore, Abdel-Rahman et al. [156] also reported that in the multi-pulse fed-batch mode, lactic acid concentration was 180.6 g·L$^{-1}$ with 0.65 g·L$^{-1}$·h$^{-1}$ volumetric productivity.

### 8.4. Continuous Fermentation Mode

In the continuous fermentation mode, fresh medium is continuously added to the fermenter while the already existing broth containing the used medium and cells is harvested at the same rate; hence, the substrates and product concentrations are constantly maintained [147]. Consumed nutrients are replaced, and toxic metabolites are removed from the culture [157]. Production of lactic acid at the rate of 1.56 g·L$^{-1}$·h$^{-1}$ by *Enterococcus faecium* in continuous fermentation was reported [158]. Ahring et al. [159] reported lactic acid production at 3.69 g·L$^{-1}$·h$^{-1}$ by *Bacillus coagulans* using continuous fermentation. The continuous fermentation done by Olszewska-Widdrat et al. [160] indicated that the highest lactic acid productivity (10.34 g·L$^{-1}$·h$^{-1}$) was achieved in the continuous mode when molasses was used. The prominent advantage of the continuous fermentation mode is the high productivity of lactic acid, and one of its disadvantages is the high risk of contamination. Several fermentation modes used in lactic acid production from different substrates using various microorganisms are listed in Table 4.

**Table 4.** Different fermentation modes in L-lactic acid production, concentrations, yield, and productivity.

| Fermentation Mode | Substrate | Microorganisms | $C_{LA}$ [c] (g·L$^{-1}$) | $P_{LA}$ [d] (g·L$^{-1}$·h$^{-1}$) | $Y_{LA}$ [e] (g·g$^{-1}$) | References |
|---|---|---|---|---|---|---|
| Fed-batch | Jerusalem artichoke tuber extract | *Lactobacillus* sp. G-02 | 141.50 | 4.70 | 0.524 | [161] |
| | | *Lactobacillus* sp. G-02 and *Aspergillus niger* [b] (mixed culture) | 120.50 | 3.34 | 1.50 | [162] |
| Fed-batch | Chicken hydrolyzate | *Lactobacillus casei* | 116.50 | 4.000 | 0.984 ± 0.10 | [151] |
| Repeated batch | Corn starch | *Rhizopus oryzae* NRRL | 98.20 [a] | 2.040 | 0.818 | [155] |
| Batch | Cheese whey | *Lactobacillus* sp. RKY2 | 94.06 | 1.060 | 0.980 | [163] |
| Batch | Jerusalem artichoke tuber extract | *Lactobacillus paracasei* KCTC 13169 | 92.50 | 1.280 | 0.980 | [164] |
| Batch (SmSF) | Paper sludge | *Rhizopus* sp. | 80.00 | 0.097 | 0.62–0.65 | [165] |
| Batch (SmSF) | Paper sludge | *Lactobacillus Rhamnosus* ATCC 7469 | 73.00 | 2.90 | 0.970 | [166] |
| Batch (SHF) | White rice bran hydrolyzate | *Lactobacillus rhamnosus* | 56.00 | 0.78 | NM | [167] |
| Repeated batch | Wood hydrolyzate | *Enterococcus faecalis* RKY1 | 48.60 | 1.40 | 0.970 ± 0.20 | [20] |
| Fed-batch (SmSF) | Cellulosic biosludge | *Lactobacillus Rhamnosus* CECT-288 | 42.00 | 0.87 | 0.378 | [168] |
| Continuous | Whey permeate | *Lactobacillus Helveticus* R211 | 42.00 | 21.00 | NM | [169] |
| Batch | Apple pomace | *Lactobacillus Rhamnosus* ATCC 9595 CECT28 | 32.50 | 5.41 | 0.880 | [170] |
| Continuous | Sago starch | *Enterococcus faecalis* | 16.60 ± 0.80 | 1.10 | 0.93 ± 0.20 | [158] |

[a]: After 48 h culture; [b]: the invertase producer; [c]: Lactic acid concentration (g·L$^{-1}$); [d]: Lactic acid productivity (g·L$^{-1}$·h$^{-1}$); [e]: Yield, the mass of lactic acid produced (g) per mass of substrate consumed (g); NM: Not Mentioned.

## 9. Lactic Acid Production Using Lignocellulosic Biomass

Sufficient literature sources are available on laboratory-scale lactic acid production using lignocellulosic materials. In lactic acid production using lignocellulosic materials, the sugar-containing hydrolyzate replaces refined sugar as the feedstock for fermentation. Different methods used to produce lactic acid have been described in the literature. These methods include simultaneous saccharification and fermentation (SmSF), separate hydrolysis and fermentation (SHF), simultaneous saccharification and co-fermentation (SSCF), separate hydrolysis and co-fermentation (SHCF), and consolidated bioprocessing (CBP) [29].

Saccharification refers to the hydrolysis of polysaccharides, e.g., lignocellulosic materials containing carbohydrates, to fermentable sugars. In liquefaction and gelatinization, heat at 90–130 °C is applied to raw carbohydrate-containing materials for 20 min. The resulting product is saccharified to produce fermentable sugar, which is used as a substrate for fermentation that yields lactic acid [171]. Simultaneous saccharification and fermentation (SmSF) combine enzymatic hydrolysis of lignocellulose and fermentation of the lignocellulosic hydrolyzate in a single step. Several studies have been done on SmSF; Marques et al. (2008) [166] reported the production of 0.97 g·g$^{-1}$ of lactic acid from recycled paper using SmSF by *Lactobacillus rhamnosus*. Likewise, Chacón et al. (2021) [172] reported the production of 5.1 g·L$^{-1}$ of lactic acid from 2% (*w/v*) municipal solid waste cellulosic pulp using SmSF by *Bacillus smithii* and 62 g·L$^{-1}$ of lactic acid from beechwood and pine was produced by *Lactobacillus delbrueckii* subsp. *bulgaricus* after 72 h of incubation using SmSF [173].

The advantages of SmSF include rapid processing time, less enzyme loading, reduced end-product inhibition of hydrolysis, high productivity, and a single reaction vessel [29,33]. SmSF is proven to be the best as it gives high substrate concentration in a low reactor volume and is cost-effective [11]. However, the major disadvantage of SmSF is the difference in enzymatic hydrolysis optimal conditions (pH < 5 at 50 °C) and optimal fermentation conditions (pH 5–7 at 37–43 °C) [30].

In the separate hydrolysis and fermentation (SHF), raw materials are first pretreated, and if lignocellulosic materials are used, the lignin is removed after pretreatment. The pretreated materials are then subjected to enzymatic hydrolysis/saccharification, followed by fermentation of the hydrolysate [174]. The drawback of SHF includes decreased lactic acid productivity due to the immense pretreatment process of lignocellulosic materials [23,147].

In general, literature on lactic acid production through co-fermentation is limited, unlike on ethanol production. In simultaneous saccharification and co-fermentation (SSCF), enzymatic hydrolysis is performed simultaneously with the co-fermentation of sugars such as glucose and xylose [175]. The advantages of SSCF include reduced capital cost [175]. The SSCF process in lactic acid production has not been extensively utilized; however, Patel et al. (2006) [38] reported simultaneous saccharification and co-fermentation of crystalline cellulose and sugarcane bagasse hemicellulose hydrolyzate resulted in 45 g·L$^{-1}$ L-lactic acid production by *Bacillus* sp. strain 36D1.

Separate hydrolysis and co-fermentation (SHCF) are similar to separate hydrolysis and fermentation (SHF), except that the substrate for fermentation contains at least two fermentable sugars such as hexose (e.g., glucose) and pentose (e.g., xylose) that are co-fermented by selected bacteria strains.

Consolidated bioprocessing (CBP) is a complex process that involves the integration of complex bioprocesses such as cellulolytic enzyme production by at least one microorganism for hydrolysis and fermentation of sugar by another microorganism in a single unit operation to produce lactic acid [176]. Consolidated bioprocessing is cost-effective as no external enzyme loading is required for the process, and there are lower contamination risks since the process is carried out under high temperatures [177]. It was reported that in CBP, *Paenibacillus macerans* IIPSP3 (MTCC 5569) hydrolyzed cellulose to glucose and fermented glucose to lactic acid under aerobic conditions with no growth inhibition in the presence of lignin [177].

Lactic acid is mostly produced from sugar-containing hydrolyzate via SHF or by a single-step of starchy or cellulosic wastes conversion using amylolytic lactic acid-producing

microorganisms (direct conversion) or via simultaneous saccharification and fermentation (SmSF) by adding enzymes and inoculum together (Figure 8) [11]. Direct conversion of starch to lactic acid using lactic acid-producing fungi is cost-effective, unlike the conversion of starch or cellulose to sugar, which consumes energy during liquefaction or saccharification [178]. Microorganisms such as *Rhizopus oryzae* and *Lactobacillus amylovorus* directly convert starch to lactic acid [138,178].

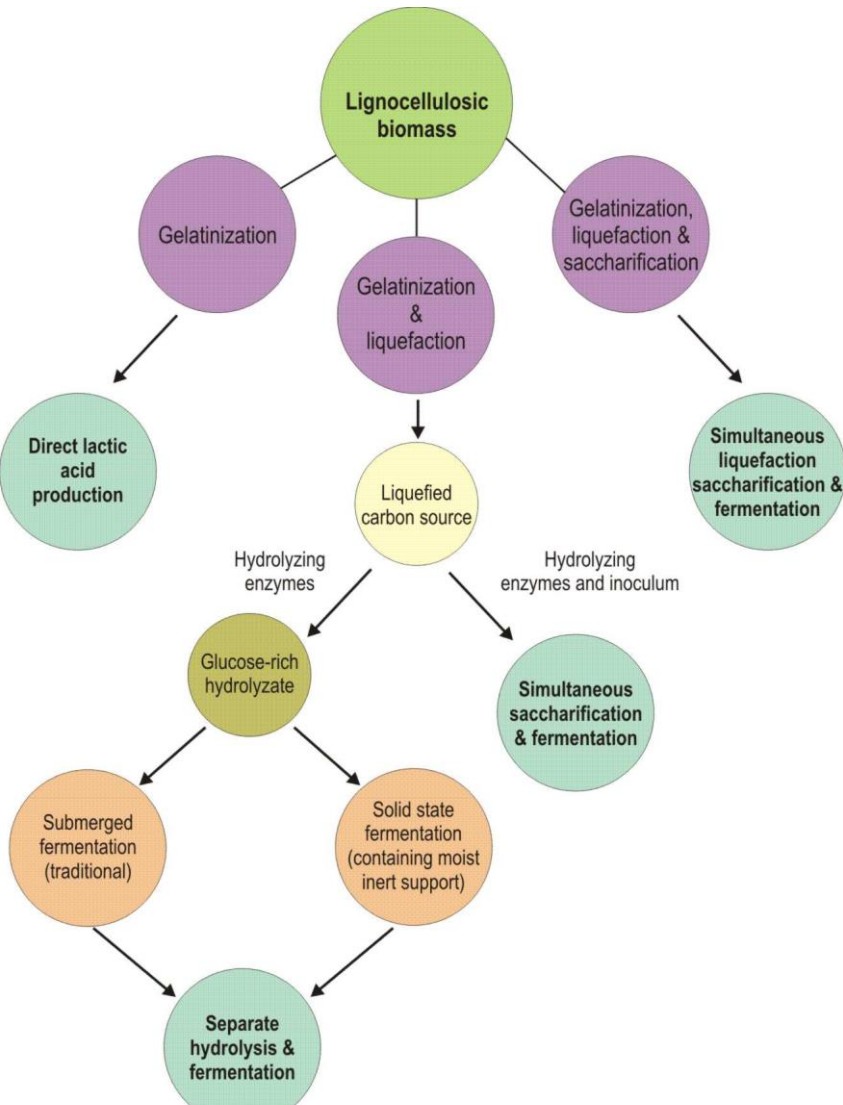

**Figure 8.** Processes in the production of lactic acid from lignocellulosic biomass (Adapted from John et al. (2007) [11]).

Coupling the enzymatic hydrolysis of carbohydrates and microbial fermentation of the derived glucose into a single step is economically attractive [179]. Microorganisms used in converting starchy biomass to lactic acid in single-step fermentation include amylolytic LAB [179,180]. The two-step process is expensive compared to the single-step process as the two-step involves enzymatic saccharification to glucose followed by glucose fermentation resulting in lactic acid production [179]. An appreciable number of studies have also been done on the application of immobilized cell systems in the optimization of the production of lactic acid [181,182], but no sufficient information is available on the industrial applications of immobilized cells in lactic acid production.

In addition to these fermentation methods, reports have shown that solid substrates such as bran, paper pulp, bagasse, etc., have been used for solid-state fermentation, and

the sugar-containing hydrolyzate has been used for submerged fermentation [137,183,184]. Solid-state fermentation (SSF) is a process carried out in a solid matrix (inert support or support/substrate) with the absence or near absence of water; however, the substrate has enough moisture to support microbial growth and their metabolic activities [185,186]. SSF mainly uses residues such as solid agro-industrial wastes, such as wheat bran, rice bran, or sugarcane bagasse, etc., as the substrate that serves as the carbon source [185].

In solid-state fermentation, substrates are utilized slowly and steadily; hence, they can be used for a long-time during fermentation [183]. Fungi and other microorganisms that require fewer moisture contents are mostly used in SSF fermentation. Lactic acid production by SSF has been reported. Solid-state fermentation was used to produce lactic acid (137 g·L$^{-1}$) at the rate of 1.38 g·L$^{-1}$·h$^{-1}$ by using *Rhizopus oryzae* [137]. In addition, Rojan et al. (2005) [178] reported the conversion of cassava bagasse by *Lactobacillus* in solid-state fermentation yielded L-lactic acid 0.58 g·g$^{-1}$ initial substrate. Submerged fermentation (SmF) uses a free-flowing liquid substrate such as broth or molasses [183]. Submerged fermentation (SmF) allows the growth of microorganisms since the broth contains nutrients, and the production of enzymes occurs in the broth as the microorganisms interact with the nutrients [183]. Substrates are utilized rapidly in SmF; as a result, nutrients need to be constantly replaced or supplemented. Submerged fermentation is conducted in a fed-batch or continuous mode.

Sugarcane bagasse used for lactic acid production by *Rhizopus oryzae* in solid-state fermentation resulted in the production of 137 g·L$^{-1}$ L-lactic acid, whereas the comparative study using submerged fermentation produced 93.8 g·L$^{-1}$ L-lactic acid [137]. Nonetheless, approximately 90% of lactic acid is produced commercially by submerged fermentation [120].

## 10. Industrial/Commercial Fermentative Lactic Acid Production and Purification

Commercial/industrial production of lactic acid involves fermentation of the fermentable sugars and the purification of the fermentation broth to obtain pure lactic acid. In general, fermentative production of lactic acid can utilize standard fermentation technology [19]. Presently, lactic acid is commercially produced from starch fermentation. However, the generation of lactic acid by fermentation of renewable agricultural feedstock resources such as lignocellulosic materials, food wastes, dairy wastes, and beverage industry wastes that contain fermentable sugars reduces the production cost, renders waste into a resource and reduces challenges in the environmental disposal of the wastes [49,187]. Little information is made available on the extensive use of lignocellulosic materials for the commercial/industrial production of lactic acid. However, Agblevor and Evans (2004) [188] presented an industrial method for producing lactic acid from agricultural livestock, specifically lignocellulosic materials such as soybean hulls. This industrial method includes size reduction by mechanical or steam explosion pretreatment, followed by hydrolysis, which may be either enzymatic or acid hydrolysis, and fermentation.

In the industrial production of lactic acid, hydrolysis, and fermentation can be conducted simultaneously or separately [188]. For instance, in the industrial production of lactic acid, enzyme hydrolysis of the pretreated material and microbial fermentation occur simultaneously in the same vessel (SmSF). Or the pretreated material is first subjected to enzymatic hydrolysis/saccharification, followed by microbial fermentation of the hydrolysate (SHF) [174,188]. It should be noted that the microorganism selected for fermentation allows stereospecific lactic acid to be produced as desired. For instance, reports have shown that homofermentative LAB such as *Lactobacillus delbrueckii* subsp. *bulgaricus*, *Lactobacillus acidophilus*, *Lactobacillus helveticus*, and fungi species from the genus *Rhizopus*, with their amylolytic enzyme activity, produce L-lactic acid, which can be used as a precursor for polylactic acid (PLA) production [20,33,46,147]. Different methods have been used in the commercial production of lactic acid; the most common methods include classical calcium lactate and ammonium lactate processes [22]. Generally, the mode used during fermentation varies from batch, fed-batch, repeated batch to continuous fermentation, etc. [22,23]. Continuous fermentation mode offers high lactic acid productivity, while fed-batch fermen-

tation mode gives high lactic acid yield. However, batch modes are commonly used in the industry for lactic acid production [19].

*10.1. Classical Calcium Lactate Process for Lactic Acid Production and Purification*

In the classical calcium lactate process of lactic acid production, a non-corrosive reactor is usually used for fermentation as the corrosion of the vessel can lead to contamination of the fermentation fluid by soluble heavy metals. In classical lactic acid manufacturing, fermentation is done in batch mode. Culture and medium containing a fermentable sugar, e.g., glucose with a concentration between 120 and 180 g·L$^{-1}$, and complex nitrogen sources (e.g., the mixture of inorganic nitrogen such as ammonium phosphate and ammonia phosphate) with complex organic materials such as yeast extracts, peptone, etc. that yields between 1 and 10 g·L$^{-1}$ are added to the reactor [22].

Fermentation is conducted in reactor volumes of more than 100 m$^3$ and at a temperature higher than 40 °C, depending on the microorganism used. For instance, if *L. delbrueckii* is used, the temperature is set up to 50 °C; possible contamination could be avoided at this temperature [22]. Agitation is done during fermentation, and calcium carbonate is added in increments or at the beginning of fermentation to maintain the pH between 5.5 and 6.0. The active fermentation is completed after 2–6 days, depending on the used-up carbon source concentration. The reaction involved in the production of calcium lactate during fermentation is in Equation (2). The produced calcium lactate determines the upper limit of sugar concentration. The calcium lactate produced is passed through the primary filter; the sludge is separated from calcium lactate, and the filtered calcium lactate goes into a decomposer tank.

$$\text{Lactic acid + Calcium carbonate} \rightarrow \text{Calcium lactate + Water + Carbon dioxide} \quad (2)$$

$$\text{Calcium lactate + Sulphuric acid} \rightarrow \text{Lactic acid + Gypsum} \quad (3)$$

$$\text{Lactic acid + Methanol} \rightarrow \text{Methyl lactate + Water} \quad (4)$$

$$\text{Methyl lactate + Water} \rightarrow \text{Lactic acid + Methanol} \quad (5)$$

The purification of calcium lactate (obtained as the fermentation product) to pure lactic acid involves three steps. In the first step, sulphuric acid reacts with calcium lactate in the decomposer tank, yielding dilute lactic acid and calcium sulfate Equation (3). The dilute lactic acid is sent to a counter-current reactive distillation column or a bubble column in the second purification step (Figure 9). It is concentrated and esterified with rising methanol in the presence of concentrated sulphuric acid (a catalyst) to produce methyl lactate and water (Equation (4)) [189,190]. The esterification reaction is depicted in Figure 9.

The reactive distillation operation composes a simultaneous reaction and separation process; as methyl lactate flows to the bottom of the column, water and the residual methanol migrate to the distillation section, where they are separated [189]. Fractional distillation was used to remove methyl lactate by-products of fermentation impurity carboxylic acid resulting in high pure methyl lactate (~98% wt.). The third step involves pure methyl lactate hydrolysis to high-purity lactic acid (Equation (5)). In this step, methyl lactate is subjected to hydrolysis using pure distilled water or de-ionized water in the presence of pure lactic acid as the auto-catalyst to avoid impurities, followed by activated carbon treatment (Figure 10).

During activated carbon treatment, carbon is pretreated with dilute, highly pure L-lactic acid solution in water. This third step results in obtaining high lactic acid purity (99% wt. on a dry basis) at an increased rate of reaction [189,190]. The desired strength of the highly pure L-lactic acid in water is obtained by concentrating pure lactic acid in the evaporator [190]. Methanol, the by-product of the hydrolysis, is recycled back to the bubble column (Figure 10), reducing energy consumption and inventory cost.

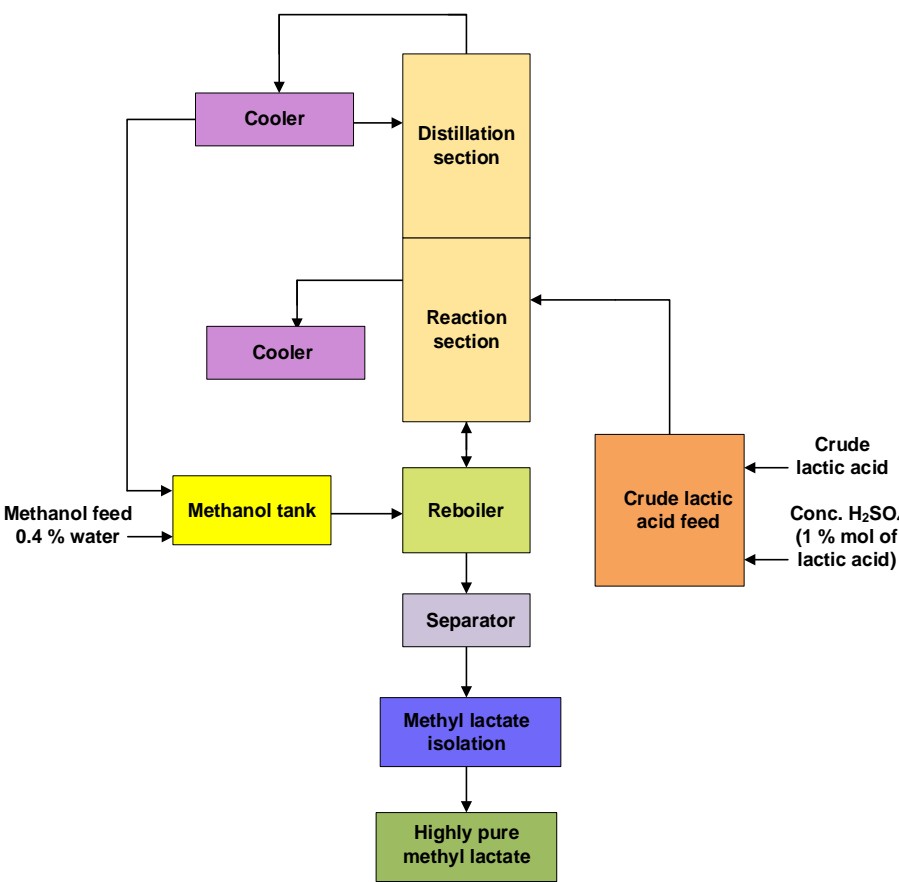

**Figure 9.** Block diagram for the second step of lactic acid purification that involves esterification of lactic acid with methanol in the presence of concentrated sulphuric acid forming methyl lactate (Adapted from Bapat et al. (2014) [189]).

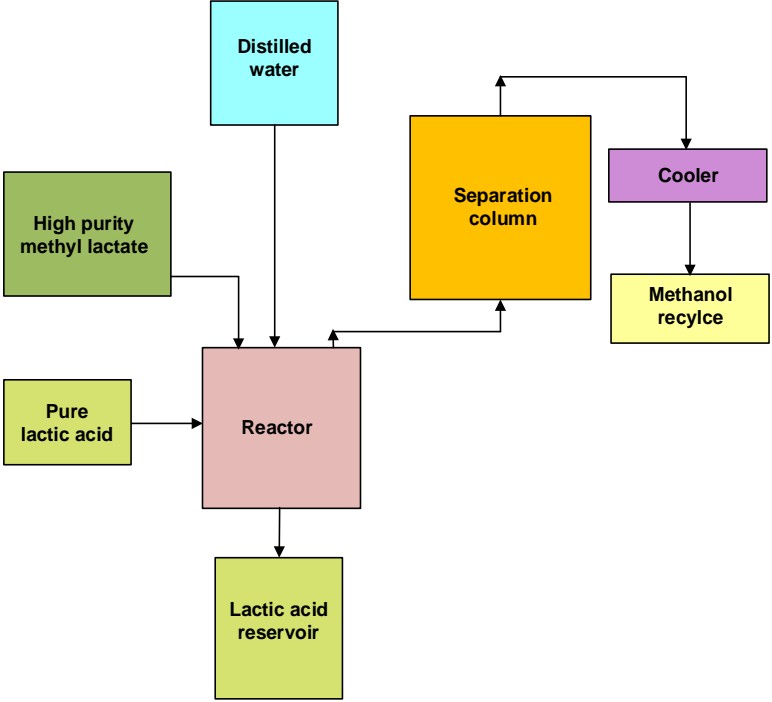

**Figure 10.** Block diagram for the third step of lactic acid purification involves hydrolysis of methyl lactate and activated carbon treatment (Adapted from Bapat et al. (2014) [189]).

*10.2. Ammonium Lactate Process in Lactic Acid Production*

The process steps in the ammonium lactate process are like that of the calcium lactate process, except that industrial fermentation is conducted using ammonia liquor. Ammonia used as a neutralizing agent in fermentative lactic acid production reduces the acidity of the fermentation broth and reacts with lactate to form ammonium lactate (Equation (6)) [22]. Lactic acid is recovered from ammonium lactate by acidulation with sulphuric acid, followed by ammonium sulfate salt crystallization (Equation (7)). Lactic acid can also be recovered from ammonium lactate by lactic acid esterification using alcohol and back recovery using pure distilled water or de-ionized water.

Sun et al. (2006) [191] reported the extraction of lactic acid from a fermentation broth by esterification and hydrolysis; in the study, butanol reacts with ammonium lactate obtained directly from fermentation for 6 h to produce butyl lactate. Sun et al. (2006) [191] further stated that the cation exchange resin used in hydrolysis was modified by replacing sulphuric acid with stannous chloride as the catalyst; neutral ammonium lactate replaced lactic acid as the starting material, and the ammonium lactate was purified. The sequential hydrolysis of purified ammonium lactate in the presence of cation exchange resin in the H$^+$ form for 4 h produced 89.7% lactic acid at a purity level of 90% [191].

$$\text{Lactic acid} + \text{Ammonia} \rightarrow \text{Ammonium lactate} \tag{6}$$

$$\text{Ammonium lactate} + \text{Sulphuric acid} \rightarrow \text{Lactic acid} + \text{Ammonia} \tag{7}$$

Although open sources provided limited data for industrial lactic acid production and purification of lactic acid [192], the separation methods widely reported include precipitation, solvent extraction, ion exchange, and membrane separation processes such as electrodialysis, reverse osmosis, and ultrafiltration [193–198]. Lactic acid can be separated from the fermentation broth by adsorption, extraction, and membrane separation, but microbial cells cannot be separated by adsorption and extraction; membrane-based separations solve this challenge [199]. In membrane separation, microfiltration membranes separate microbial cells or suspended colloidal particles [194,197], and in the continuous fermentation mode, the filtered microbial cells are subsequently recycled into the bioreactor (fermenter), resulting in high cell concentration and high lactic acid productivity. The ultrafiltration membranes retain proteins and microbial cells [200], while nanofiltration membranes separate cells, proteins, salts, nutrients, and unconverted carbohydrates from lactic acid. Reverse osmosis, or nonporous membrane based on solution diffusion mechanism, separates the same components as nanofiltration membranes but at high pressure.

## 11. Challenges in LA Production from Lignocellulosic Materials

Agro-wastes such as lignocellulosic materials and food waste as feedstocks in LA production reduce the fermentation cost and offer environmental preservation and sustainability. However, biological LA production from lignocellulosic materials may be challenging because most microbes cannot directly metabolize lignocellulose. Problems in LA production from lignocellulosic materials include inhibitory by-products formed during pretreatments, substrate-, feedback- and end-product inhibition and separation and purification.

*11.1. By-Products Formed in the Delignification of Lignocellulose during Pretreatments*

Different types of by-products are formed during the pretreatment of lignocellulose, and the type of by-product formed depends on the pretreatment method. Some hemicelluloses contain arabino-4-*O*-methylglucurono-D-xylan and *O*-acetyl-galactoglucomannans or arabino-(*O*-acetyl-4-*O*-methylglucurono)-D-xylans with p-coumaric and ferulic acid and lignin contains phenylpropanoid units [201,202]. During hemicellulose hydrolysis, pentose or hexose, uronic acid, acetic acid, and phenolic acids are formed [203]. In acid hydrolysis, pentoses and uronic acid undergo dehydration to form 2-furaldehyde, and hexoses are dehydrated to 5-hydroxymethyl-2-furaldehyde (HMF). 5-hydroxymethyl-2-furaldehyde

further degraded to levulinic acid and formic acid when the temperature, reaction time, and acid concentration were increased [204–206]. In addition, in acid hydrolysis, phenolic compounds are formed during the splitting of ß-O-4-ether and other acid-sensitive linkages; lignin and acetic acid are formed from hemicellulose acetyl groups [75,207].

Alkaline pretreatments lead to polysaccharide degradation, forming saccharinic acid, formic acid, acetic acid, phenolic compounds, hydroxy acid, and dicarboxylic acid [208]. Under oxidative conditions, phenolic compounds are oxidized to carboxylic acids and 2-furaldehydes or furfural to furoic acid [75,209]. Some generated by-products could adversely affect the downstream processes. For instance, inhibitors such as acetic acid, formic acid, levulinic acid, aldehydes, and some phenolic compounds formed during delignification negatively affect microbial growth, substrate utilization by microorganisms, and fermentation [210,211]. Specifically, formic acid migrates into the cell membrane and hinders microbial activities. The combination of acetic acid, formic acid, glycolic acid, coumaric acid, and phenolics inhibited *Lactobacillus lactis* growth, while levulinic acid at a concentration of 10 g·L$^{-1}$ inhibited *Bacillus smithii* growth [212]. Aldehydes are toxic; they cause plasma membrane damage, inhibit microbial growth, and directly inhibit glycolysis and fermentation [210,213]. Phenolic compounds damage microorganisms' cell membranes resulting in the leakage of intracellular components [214]. Phenolic compounds could also impair microorganisms' growth [215].

### 11.2. Feedback-, Substrate- and End-Product Inhibition

During enzymatic hydrolysis of lignocellulose hydrolysate, there could be feedback inhibition where the increased cellobiose and glucose concentrations inhibit exoglucanase (CBH) and endoglucanase (EG) activities of cellulases that break down cellulose to form cellobiose and, finally, glucose. The cellulase activity inhibition usually decreases the sugar formation rate [33], whereas substrate inhibition occurs during fermentation when the high concentration of glucose or pentose inhibits the growth of lactic acid-producing microorganisms due to decreased water activity, increased osmotic pressure, and lysis of cells [216,217]. End-product inhibition occurs due to the overexposure of lactic acid-producing microorganisms to lactic acid during fermentation [6]. Long-time exposure to these microorganisms causes lactic acid to penetrate their cell membranes, resulting in increased intracellular LA and cell disruption due to the change in membrane potential, thereby causing cell death [218].

### 11.3. Separation and Purification Challenges

Some of the separation methods have some drawbacks. For instance, the purification of lactic acid produced in a classical way involves several downstream treatment steps, such as precipitation, conventional filtration, acidification, distillation, carbon adsorption, and evaporation, to mention a few [199]. The high cost of reagents used during precipitation, the generation of a large quantity of wastewater, and the production of gypsum are the major drawbacks of using precipitation methods for lactic acid purification. The high cost of membranes limits the use of membrane processes in lactic acid separation and purification. In solvent extraction, extractants are toxic to microorganisms in in-situ extractive fermentation [219]; the high exchange area required for adequate separation results in high equipment costs and solvent recovery.

## 12. Strategies to Circumvent Difficulties in the LA Production from Lignocellulose

Different strategies have been used to reduce substrate inhibition. The fed-batch culture approach in fermentation increased maximum viable cell concentration and prolonged culture lifespan; still, unstable substrate concentration that stresses the LA-producing microorganism makes the fed-batch culture unsuitable for overcoming substrate inhibition [33]. Reports have shown that continuous and semi-continuous fermentation processes in LA production could reduce substrate inhibition [220]. Lactic acid must be removed from the fermentation broth to reduce end-product (lactic acid) inhibitory effects. Different approaches

used to remove lactic acid from the fermentation broth include electrodialysis [221], nanofiltration [222,223], an ion-exchange resin [224], and extraction from the fermentation [225]. Reports have shown that continuous removal of lactic acid by electrodialysis or extraction resulted in higher lactic acid concentration and yield [226,227]. In addition, the expression of LA-producing enzymes in fungi, e.g., yeast, could overcome end-product inhibition [228].

Several methods developed to minimize feedback inhibition include the removal of sugar during hydrolysis by ultrafiltration or simultaneous saccharification and ferment on (SmSF) [229,230], optimizing cellulase conditions during hydrolysis [231], avoiding cellobiose accumulation by supplying ß-glucosidase and improving ß-glucosidase activity in the cellulose [232,233]. Different strategies used to detoxify lignocellulose hydrolyzate by-product inhibitory effects include evaporation and membrane filtration (physical methods). Volatile inhibitors such as acetic acid and furfural can be evaporated under vacuum conditions. Wickramasinghe and Grzenia (2008) [234] used the adsorptive membrane to remove acetic acid from lignocellulosic hydrolyzate. Fayet et al. (2018) [235] found the DK membrane most suitable for removing wheat straw hydrolyzate inhibitors. Nanofiltration with a reverse osmosis system was used to remove lignocellulosic hydrolyzate inhibitors and concentrate sugar [236]. Another approach used to remove by-product inhibitors is by chemical processes involving alkaline detoxification, ion exchange, and wood charcoal (biochar) adsorption [237–239]. The acidic lignocellulosic hydrolyzate can be neutralized by adding alkali solutions such as calcium hydroxide, ammonium hydroxide, or sodium hydroxide. Guo et al. [238] reported calcium hydroxide as the most effective in removing total phenols. Ahmed et al. [237] used dry calcium carbonate as an acid-neutralizing agent to mitigate handling and phenolic compounds problems. Lee and Park [239] used activated charcoal to detoxify lignocellulosic hydrolyzate inhibitors in a fixed bed column containing biochar; 99% of furfural and other phenolic compounds were removed from dilute acid-pretreated biomass hydrolyzate.

Biological detoxification and removal of inhibitors using microorganisms and their enzymes are eco-friendly but may be time-consuming. Suman et al. [240] reported 66% removal of inhibitors such as phenolics from the sugarcane bagasse prehydrolyzate by *Trametes maxima* IIPLC-32 laccase. *Bordetella* sp. completely removed furfural, 94% HMF, and 82% acetic acid from sugarcane bagasse hydrolyzate within 16 h [241].

Genetic engineering has been used to develop recombinant microorganisms to improve their resistance to formic acid, acetic acid, furfural, and phenolics to alleviate the inhibitory by-product effects [73,242–244]. The engineered furfural resistance of *Escherichia coli* resulted in improved resistance to sugarcane bagasse hydrolysate [243], while the engineering of *S. cerevisiae* for improved resistance to formic acid and acetic acid enhanced formate dehydrogenase and transaldolase activity [244].

## 13. A Proposed Model for LA Production and Purification: Enzyme and Cell Recycling Continuous Simultaneous Saccharification and Fermentation

Alleviating problems using lignocellulose as the substrate for LA production have been investigated extensively [73,147,220,228,245]. A cost-effective model that will overcome or reduce the challenges associated with LA production using lignocellulosic materials is proposed. This model comprises a feed vessel of media/nutrients, a bioreactor containing lignocellulose hydrolyzate for saccharification, substrate (e.g., glucose obtained after enzymatic hydrolysis), enzyme(s), inoculum and media for fermentation, an ultrafiltration membrane system for enzyme and cell recycling, a granulated active charcoal column (for decolorization), a chelating resin column (for multivalent metal ions removal), a two-stage electrodialysis (convectional electrodialysis and bipolar electrodialysis) and a permeate vessel (the purified lactic acid vessel). The simultaneous saccharification and fermentation (SmSF) method will be employed, where saccharification/enzyme hydrolysis and fermentation occur in one vessel. The hydrolyzing enzyme, e.g., cellulase and the inoculum, preferably the homofermentative L-lactic acid-producing bacterium, will be used in the saccharification and fermentation of the selected cellulosic wastes. The challenge in using lactic

acid-producing bacteria is their mesophilic nature since saccharification is usually done at a higher temperature. Studies have shown that this challenge can be circumvented by conducting SmSF at a mesophilic temperature with an increased saccharification time [246]. The SmSF anaerobic system will be operated in a continuous mode with sequential input that will supply fresh media/nutrients into the bioreactor and continuous ultrafiltration of enzyme and cell recycling into the bioreactor. Prior to the two-stage electrodialysis, the fermentation broth containing lactate salt with multi-metal ions will be pretreated by passing it through the granulated active charcoal column (for decolorization) and then to the chelating resin column (for multivalent metal ions removal) [247]. The lactate salt will undergo convectional (monopolar) electrodialysis followed by bipolar electrodialysis to obtain pure lactic acid (Figure 11).

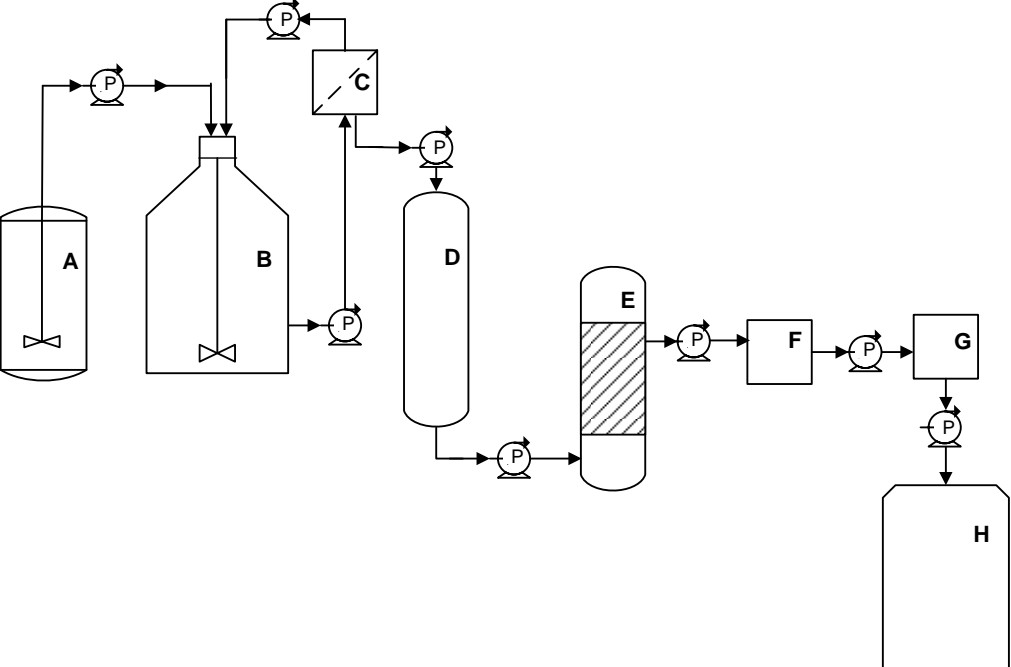

**Figure 11.** The schematic diagram of continuous lactic acid production with enzyme and cell recycling by ultrafiltration coupled with granulated active charcoal and chelating resin columns and a two-stage electrodialysis. A: Feed vessel containing fresh media/nutrients; B: Anaerobic bioreactor containing lignocellulose hydrolyzate (before enzymatic hydrolysis), substrate (e.g., glucose obtained after enzymatic hydrolysis), enzyme(s), inoculum and media; C: Ultrafiltration membrane; D: Granulated active charcoal column; E: Chelating resin column; F: Monopolar (convectional) electrodialysis; G: Bipolar electrodialysis; H: Permeate (lactic acid) vessel; and P: low/medium/high pump operated at different flow rates.

In SmSF, the simplest sugar formed during hydrolysis is immediately fermented into LA; this prevents sugar accumulation that may result in feedback and substrate inhibition. The proposed membrane enzyme and cell recycling system through ultrafiltration allows the recycling of enzymes for reuse in several batches in the hydrolysis of hydrolyzate, which reduces the cost and quantity of enzymes needed for hydrolysis. The inoculation of the used cells (biomass) in a continuous cycle of the fresh medium increases the cell concentration in the bioreactor [154]. Continuous fermentation with cell recycling results in high LA productivity, yield, and concentration as it eliminates end-product inhibition [248–250]. Adding alkaline to maintain near-neutral pH for fermentation results in lactate salt production instead of lactic acid.

However, a two-stage electrodialysis system overcomes salt and gypsum disposal problems. In the two-stage electrodialysis, the convectional (monopolar) electrodialysis separates and concentrates lactate salt, and bipolar electrodialysis (water-splitting electro-

dialysis unit with the bipolar membrane) converts lactate salt to lactic acid [247,251]. The pretreatment of fermentation broth containing lactate salt with multi-metal ions before electrodialysis prevents the decrease in the electrodialysis efficiency due to dye fixing on the membrane and irreversible damage of electrodialysis membranes, especially the bipolar electrodialysis.

## 14. Conclusions

Lactic acid is used on a large scale in the food, chemical, pharmaceutical, biomedical, and cosmetic industries. It is an essential precursor for synthesizing polylactic acid, a biodegradable polymer that replaces petroleum-based plastics. Lactic acid production via the chemical route is not cost-effective. Therefore, a biological process is used in the commercial or industrial production of lactic acid. The microorganism selected for fermentation allows the desired production of stereospecific lactic acid (L- or D-lactic acid). There are more movements toward using complex polymeric substrates such as lignocellulosic materials, food and dairy wastes, and beverage industry wastes in lactic acid production because refined sugars, the simple carbon sources for microorganisms to ferment, are expensive. However, these complex polymeric substrates are pretreated to make the carbon source accessible to the microorganisms. Lignocellulose pretreatments may produce inhibitors that could lead to a substrate- or feedback inhibition. Different strategies are employed in detoxifying lignocellulosic hydrolyzate. The microorganisms that produce lactic acid from the hydrolyzed pretreated lignocellulose include various genera of bacteria and fungi, and these can be genetically modified. Different fermentation modes are available for lactic acid production, but a batch fermentation mode is extensively used in the industrial production of lactic acid. Nonetheless, fed-batch and continuous fermentation modes are more advantageous because the controlled substrate enhances microbial growth in the fed-batch mode, leading to higher lactic acid yield. In the continuous mode, the constant maintenance of substrates and product concentrations by continuously adding fresh media to the fermenter and removing the already existing broth containing lactic acid and cells at the same rate results in high lactic acid productivity and overcomes low pH challenges that result in end-product inhibition. Precipitation, solvent extraction, adsorption, filtration, distillation, and membrane separation are used in lactic acid separation. However, the precipitation method produces calcium sulfate (gypsum), which could lead to environmental contamination due to large volumes being produced. At the same time, adsorption and extraction-based processes need additional separation of microbial cells. Membrane-based separation processes offer the best lactic acid separation and purification. Only classical calcium lactate processes in the batch mode have been mainly used in the industrial production of lactic acid, and acidification of calcium lactate using sulphuric acid produces lactic acid and calcium sulfate (gypsum) as the by-product. Due to challenges in lactic acid production from lignocellulosic materials, we thereby propose a promising option for the industrial production of LA that consists of continuous simultaneous saccharification and fermentation with an ultrafiltration membrane system for enzyme and cell recycling, a two-stage pretreatment of the lactate salt (a granulated active charcoal column (for decolorization) and a chelating resin column (for multivalent metal ions removal)), a two-stage electrodialysis (convectional (monopolar) electrodialysis and bipolar electrodialysis) and a permeate vessel (the purified lactic acid vessel).

**Author Contributions:** A.O.O. and O.d.S. contributed to the development and writing of this article. All authors have read and agreed to the published version of the manuscript.

**Funding:** This research did not receive any specific grant from funding agencies.

**Data Availability Statement:** Not applicable.

**Conflicts of Interest:** The authors declare no conflict of interest.

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
