# Peer review of "Lactic Acid: A Comprehensive Review of Production to Purification"

_processes, doi:10.3390/pr11030688_

Round 1

Reviewer 1 Report

The work entitled "Lactic Acid: A Comprehensive Review of Production to Purification" is a detailed review of this important product, of all the aspects that have to do with the production of this acid, from the upstream stages to the downstream. Therefore, it is an essential contribution for all those readers who want a general overview of how this product, vital for the food, cosmetic and pharmaceutical industries, is produced and purified. 

However, despite the merits of the work, some non-correctable shortcomings can be pointed out, such as: 

  1. A poor approach to issues related to the current and prospective market for this product, 
  2. An insufficient approach related to economic analysis and elements of profitability among the various technologies mentioned,
  3. Other applications of lactic acid, in addition to the aforementioned of being the precursor for obtaining polylactic acid (PLA). 

Additionally, it is also suggested to amend some other minor insufficiencies, and which are indicated below:

  1. The Abstract has 235 words, and according to the editorial norms, it should not exceed 200. Please, comply with the journal's standards by shortening the Abstract. 
  2. Table 1 (line 63) says in the first column: "The liquid density of aqueous solution...", where they put the values without the corresponding units. It is suggested to substitute something more concise like:

Density of aqueous solution at 25°C for

20.0% wt. ......... 1.057 g/cm3

86.6% wt. ......... 1.201 g/cm3 

  1. The values of the boiling and melting temperatures must be accompanied by their units (°C) without the parentheses that are placed. 
  2. The order of the letters for the footnotes of Table 2 must be fixed. It must be placed in order of appearance, from top to bottom and from left to right.
  3. Change from "g.L-1" to "g·L-1"; note that the "·" stay in the centre.
  4. Change from "g.L-1.h-1" to "g·L-1·h-1"; note that the "·" stay in the centre.
  5. The correct symbol of temperature in Celsius degree is "°C".
  6. For the figures from 13 to 16, I strongly recommend you redo them again. Please, consider using drawing images of equipment, such as tanks, mixers, reactors, separators, pumps, valves, etc., such as those provided by MS Visio or WondershareEDraxMax, for example. Even more, the process diagrams must be of the highest possible quality for the journal "Processes",
  7. Other comments were made directly in the manuscript.

Reviewer 2 Report

This white paper focuses on lactic acid technology, which is growing in demand due to, among other applications, the growing demand for biodegradable materials. Lactic acid can be transformed into polylactic acid, which is a plastic mass and is completely and easily biodegradable.

The content of the abstract, the set of keywords and the content of the article itself are fully consistent with the topic of Processes.

The article is a voluminous and detailed study of the state of scientific developments in the technology for the production and purification of lactic acid from lignocellulosic raw materials at the moment. The choice of lignocellulosic raw materials is absolutely justified and reflects current scientific trends: The total cost of production of lactic acid depends on the initial raw materials, since these materials account for 40-70% of the total production cost. It is recognized worldwide that the use of refined sugars for microbial transformation into lactic acid is very expensive and impractical. Agricultural waste and by-products of the food industry are currently being considered as alternative sources of raw materials. Lignocellulosic biomass is the preferred substrate because, as a raw material, it is able to meet the huge demand for lactic acid.

The article considers in detail the types of lignocellulosic raw materials and methods of its pre-treatment. The presentation of the material is presented logically, consistently and accessible, which allows you to get a fairly complete picture of the issue under consideration.

As an optional recommendation, the following considerations can be made: to create a more complete picture of possible sources of lignocellulosic raw materials, include the following information in the review:

- obtaining lactic acid from wheat straw using the autohydrolysis process is described in the work of Verkhoturova EV, Lozovaya TS, Evstaf'ev SN. Biosynthesis of lactic acid via fermentation of wheat straw autohydrolysis products. Izvestiya Vuzov. Priklad-naya Khimiya i Biotechnologiya = Proceedings of Universities. Applied Chemistry and Biotechnology. 2016; 6(3):36–41. (In Russian) https://doi.org/10.21285/2227-2925-2016-6-3-36–41.;

- to obtain lactic acid, beech and pine wood were selected as a substrate, which were previously treated by an oxidative-organo-solvent method, and the technology of simultaneous saccharification and fermentation was applied [Karnaouri A, Asimakopoulou G, Kalogi-annis KG, Lappas A, Topakas E. Efficient D -lactic acid production by Lactobacillus delbrueckii subsp. bulgaricus through conversion of organosolv pretreated lignocellulosic biomass. Biomass and bioenergy. 2020;140:105672. https://doi.org/10.1016/j.biombioe.20 20.105672];

- oat husk - agricultural waste, which can be used after pre-treatment and enzymatic hydrolysis for the biosynthesis of lactic acid [Shavyrkina NA, Skiba EA. Obtaining lactic acid from oat husks. Izvestiya Vuzov. Prikladnaya Khimiya i Biotechnologiya = Proceedings of Universities. Applied Chemistry and Biotechnology. 2021;11(1):99–106. (In Russian). https://doi.org/10.21285/2227-2925-2021-11-1-99-106].

As a note:

check the correctness of the design of primary sources so that readers can easily obtain information.

Round 2

Reviewer 1 Report

Most of my suggestions were accepted, and the corrected manuscript can be published.